# Multifocal Gastrointestinal Stromal Tumors (GISTs) of the Small Intestine in Patients with Neurofibromatosis Type 1 (NF-1): Meta-Analysis and Systematic Review of the Literature

**DOI:** 10.3390/cancers17121934

**Published:** 2025-06-10

**Authors:** Stylianos Mantalovas, Vasileios Alexandros Karakousis, Christina Sevva, Panagiota Roulia, Stavros Savvakis, Georgios Anthimidis, Konstantinos Papadopoulos, Vasiliki Magra, Nikolaos Varsamis, Christoforos S. Kosmidis, Isaak Kesisoglou

**Affiliations:** 13rd Surgical Department, AHEPA University Hospital of Thessaloniki, School of Medicine, Aristotle University of Thessaloniki, 1st St. Kiriakidi Street, 54621 Thessaloniki, Greece; steliosmantalobas@yahoo.gr (S.M.); christina.sevva@gmail.com (C.S.); panagiotar96@gmail.com (P.R.); savvstav@auth.gr (S.S.); anthimid@gmail.com (G.A.); kostaspap1995@hotmail.com (K.P.); valia.magra@gmail.com (V.M.); nikvar83@gmail.com (N.V.); kosmidisc@auth.gr (C.S.K.); ikesis@auth.gr (I.K.); 2Plastic Surgery Department, Papageorgiou General Hospital of Thessaloniki, School of Medicine, Aristotle University of Thessaloniki, Ring Road, 56403 Thessaloniki, Greece

**Keywords:** small intestine, GIST, N-F1, multifocal, solitary

## Abstract

This systematic review and meta-analysis evaluates small intestinal gastrointestinal stromal tumor (SI-GIST) prevalence patterns in neurofibromatosis type I (NF-1) patients, offering the first focused assessment of this specific anatomical subset. By isolating small intestinal cases, this work provides precise epidemiological data comparing NF-1-associated and sporadic GIST frequencies, while analyzing multifocal versus solitary tumor distributions. The study addresses key knowledge gaps regarding SI-GIST prevalence in NF-1, clinical presentation patterns, and comparative population-level risks, aiming to establish clearer diagnostic benchmarks and characterize the epidemiological burden of NF-1-associated GISTs. Insights from this study could improve clinical surveillance through evidence-based risk stratification, while the methodological focus on small intestinal tumors ensures biologically relevant comparisons. The results inform both NF-1 management guidelines and fundamental research into GIST development mechanisms, particularly regarding NF-1-related tumorigenesis pathways. This work bridges clinical epidemiology and molecular oncology by contextualizing observed prevalence patterns within current pathogenic models of GIST formation.

## 1. Introduction

Gastrointestinal stromal tumors (GISTs) are a distinct category of tumors originating from the interstitial cells of Cajal (ICCs) [1]. The pathogenic mechanism behind their development is complex. At the molecular epicenter, gain-of-function mutations in *KIT* (proto-oncogene, receptor tyrosine kinase) or PDGFRA (Platelet-Derived Growth Factor Receptor Alpha) ignite constitutive tyrosine kinase activation, propelling interstitial cells of Cajal toward neoplastic transformation through unchecked proliferative signaling cascades. NF-1-associated GISTs orchestrate their tumorigenic symphony via neurofibromin loss, unleashing RAS/MAPK (Mitogen-Activated Protein Kinase) pathway hyperactivity while conspicuously lacking the canonical KIT/PDGFRA driver mutations. The succinate dehydrogenase (SDH)-deficient variant emerges as a distinct epigenetic maverick, where SDH complex aberrations induce a CpG island methylator phenotype (CIMP), rewriting the transcriptional landscape of transformed precursors [2,3]. A major breakthrough in understanding the molecular mechanisms involved in their formation was made by Hirota et al. in 1998 [4]. In this study, it was found that the receptor for the stem cell growth factor (KIT) is expressed in the vast majority of GISTs [4]. Wild-type renegades—devoid of KIT/PDGFRA/RAS/SDH lesions—chart alternative oncogenic routes through BRAF (B-Raf proto-oncogene, serine/threonine protein kinase) mutations or noncoding RNA networks that hijack cellular homeostasis. This molecular taxonomy directly dictates therapeutic vulnerability: while KIT-mutated GISTs bow to tyrosine kinase inhibition, their SDH-deficient and NF-1-driven counterparts mount intrinsic resistance, demanding alternative therapeutic strategies [2,3]. Overall, 85% of GISTs are caused by mutations in the KIT/PDGFRA genes, while the remaining 15% result from other pathogenic mechanisms unrelated to KIT [5]. Only 1–2% of all GIST cases occur in patients carrying a mutation in the NF-1 gene, which is responsible for neurofibromatosis [6,7]. This subset of patients does not present mutations in the KIT gene and falls under the 15% of cases mentioned above [5]. This category includes cases with a deficiency in the enzyme SDH, cases with BRAF gene mutations, and, of course, cases of patients with NF-1 gene mutations and dysfunction of the corresponding protein neurofibromin, as previously mentioned [8].

Neurofibromatosis type 1 is a genetic disorder with a prevalence of 1 in 3000–3500 individuals [9,10]. It was first described in great detail by Friedrich Daniel von Recklinghausen in 1882 [11], and in many references, it bears his name [12]. It is inherited in an autosomal dominant manner [13,14]. Signs and symptoms are often present at birth, but they usually manifest before the age of 10 [13,14]. The clinical presentation of affected individuals includes café-au-lait spots, neurofibromas, Lisch nodules, scoliosis, small tumors in the iris, and distant tumors throughout the body, such as pheochromocytomas or GISTs [13,14]. Learning disabilities are also observed in young patients [15]. Diagnosis is based on clinical criteria [13,14]. These individuals often undergo surgical procedures to remove tumors from various parts of the body [16,17]. However, in most cases, patients with NF-1 experience moderate symptoms and lead a normal and productive life [18,19,20,21]. With advances in managing these patients, life expectancy has significantly increased, and in some cases, it can approach that of the general population [22,23]. This, combined with molecular testing and improved patient monitoring, has led to an increasing number of recorded cases of neurofibromatosis in the general population. Thus, the condition appears to be much more common than previously believed [13,24,25].

The dysfunction of the neurofibromin protein negatively regulates the activity of the Ras/MAPK (Mitogen-Activated Protein Kinase) molecular pathway [26,27]. This indirectly leads to the growth signaling of KIT/PDGFRA mRNA, resulting in the formation of different types of tumors each time [28,29]. GISTs represent a histological tumor category that occurs with increased frequency in patients with NF-1 and, more specifically, is observed in the small intestine [6,7,30,31,32,33]. In most cases, they are multifocal [32,33]. The clinical presentation of gastrointestinal GISTs is similar to that of any other gastrointestinal tumor [34]. They commonly manifest with bleeding, obstruction, and, less frequently, perforation. However, they are rarely the first manifestation of NF-1 [34,35,36]. A major challenge for clinicians is the bleeding caused by multifocal small intestine GISTs [36,37,38,39,40,41]. The particular difficulty lies in the localization study of the exact bleeding site, which may even necessitate intraoperative endoscopy when imaging modalities fail [42,43,44,45,46,47].

A vast number of studies attempt to examine the increased frequency of GISTs in NF-1 patients at both epidemiological and molecular levels [6,7,32,33]. Since the number of cases is small and research spans a long period, comparative analysis with the general population (i.e., sporadic GIST cases) is usually not performed. Instead, emphasis is placed on the increased prevalence of GISTs in NF-1 patients and the pathogenic mechanism of GISTs [6,7,32,33]. Additionally, studies document the various gastrointestinal sites where GISTs occur without focusing particularly on the most common site, which is the small intestine in NF-1 patients [6,7,32,33].

In the present study, we conduct a systematic review of the literature. The prevalence of small intestinal GISTs in NF-1 patients is analyzed. A statistical correlation is drawn between multifocal and solitary small intestinal GIST cases. Additionally, the prevalence of sporadic small intestinal GISTs in the general population is examined. Our findings indicate that the occurrence of small intestinal GISTs in NF-1 patients is significantly higher compared to sporadic small intestinal GISTs. As far as we could determine, this is the first systematic review with a meta-analysis focusing specifically on small intestinal GISTs. We considered it appropriate to isolate cases affecting a specific part of the gastrointestinal tract, the small intestine in this case, to make the systematic review more focused. Such an approach has greater clinical relevance and offers a more epidemiologically robust analysis, as the tumor’s pathogenesis differs. If research specifically targets the small intestine, the tendency of NF-1 patients to develop GISTs appears to be much higher than previously believed. This finding is analyzed in detail both at the epidemiological and pathogenic levels.

## 2. Materials and Methods

### 2.1. Study Design and Inclusion/Exclusion Criteria

This systematic review and corresponding meta-analysis were conducted following the PRISMA (Preferred Reporting Items for Systematic reviews and Meta-Analyses) algorithm, based on guidelines for systematic reviews (Figure 1) [48]. The study was prospectively registered in the PROSPERO database and received the ID number CRD420251002384. The entire study was based on the following PICO research question:Patients: Patients with SI (small intestinal) GISTs.Intervention: NF-1 geneComparison groups: Multifocal SI-GIST NF-1 vs. sporadic/solitary SI-GIST NF-1.Outcomes: Prevalence of multifocal SI-GIST NF-1 vs. prevalence of sporadic/solitary SI-GIST NF-1 (long period of research).

Prevalence was chosen as the primary metric for comparing the two population groups since incidence data are not accurately reported in the literature, particularly for GISTs in NF-1 patients. The two population groups differ in life expectancy and the age range in which GISTs appear. Additionally, the studies were conducted over an extended period, making it difficult to precisely calculate the overall incidence between the two groups. Prevalence serves as an indirect indicator of the tendency to develop GISTs in the two populations. All studies included in this systematic review were retrospective, as the number of NF-1 patients with GISTs is very small. For this reason, the studies were also non-randomized. The inclusion criteria required that all studies had a large patient cohort with clearly defined quantitative data for both NF-1 and GISTs in the small intestine. The exclusion criteria were as follows:Non-relevant articles.Studies that did not report quantitative data for NF-1 and GISTs.Animal studies.Case reports or systematic reviews based on case reports.Letters to the editor.Irrelevant systematic reviews.Studies with only abstracts available or those for which the full article could not be accessed.

### 2.2. Literature Search Strategy

Eligible studies included in the meta-analysis were retrieved using three search engines: PubMed, Scopus, and ResearchGate. The search algorithm used was as follows: ((NF-1) OR (GIST)) AND (GIST on NF-1 patients) OR (complications NF-1) OR (prevalence NF-1) OR (prevalence GIST) OR (GIST general population). No specific filters were applied in our search. Two scholars worked independently to choose the studies. A consensus was reached with a third researcher to settle any disputes. Additionally, a thorough examination of the reference lists of the selected studies was performed to identify potentially relevant studies using the snowball effect methodology [49].

### 2.3. Data Extraction and Assessment of the Risk of Bias

Two independent researchers collected the data and recorded it in two separate Microsoft Excel^®^ spreadsheets (Table 1 and Table 2). Both researchers analyzed GIST cases in NF-1 patients as well as GIST cases in the general population. Only small intestinal GISTs were recorded. Regarding GISTs in NF-1 patients, a detailed breakdown was performed, categorizing tumors as multifocal, solitary, or duodenal GISTs. Particular emphasis was placed not only on the tumor localization but also on the clinicopathological characteristics, such as the interstitial cell of Cajal (ICC) hyperplasia. Since all studies included in this systematic review were non-randomized, the ROBINS-E (Risk of Bias in Non-Randomized Studies of Exposures) tool was used to evaluate the risk of bias [50,51]. Articles were classified into four categories; low risk of bias, moderate risk of bias, high risk of bias, and very high risk of bias. Additionally, the overall risk of bias was assessed using the ROBINS-E algorithm, which is widely applied in epidemiological studies [50,51]. A panel was assembled to assess the bias-related questions (Figure 2), and a percentage analysis of these assessments is presented in Figure 3. The included studies represent a large and diverse patient population, covering various NF-1 research centers worldwide and different time periods. The likelihood of case duplication or overlap was considered highly improbable [50,51].

### 2.4. Data Pooling

Data were recorded in Microsoft Excel^®^ tables, using absolute numbers as reported in each study. The two independent researchers responsible for data collection cross-checked the final studies and initially compiled their datasets separately. In rare instances of discrepancies, a third researcher participated in the process to ensure data accuracy and finalize the records. A consensus was always reached within the research team before a unified dataset was produced. This approach was applied to both NF-1-associated GIST data and sporadic GIST data in the general population.

### 2.5. Meta-Analysis

The meta-analysis of GIST prevalence rates was conducted using Jamovi^®^ software (Version: 2.3.16.0), which provided pooled estimates and generated forest plots [91]. All estimates were based on the frequency data reported in the primary analyses of the included studies. For the analysis, a random-effects model was employed using the inverse variance method. The level of statistical significance was set at 0.05 in all cases. Heterogeneity among studies was assessed using Cochran’s Q statistic and quantified using the I^2^ statistic [92]. Studies were considered to have significant heterogeneity if *p* < 0.1 or I^2^ > 50%, and a funnel plot was used to assess publication bias [91,92]. Moreover, a subgroup analysis was conducted, particularly focusing on NF-1 patient data (Figure 4).

## 3. Results

### 3.1. Data Analysis

#### 3.1.1. Dataset 1: GISTs and NF-1 in the Total Number of Patients with NF-1

A total of 10 studies were included in the analysis, comprising 1386 recorded cases of NF-1 patients [53,54,55,56,57,58,60,61,62,63]. The analysis for the pooled estimate of the proportion of small intestinal GIST cases among the total NF-1 patient population per study was based on random-effects models and the inverse variance method after logarithmic transformation. The pooled proportion estimate is 0.03 with a 95% confidence interval (0.02–0.05). The values for each study are presented in Table 3 and are graphically represented in the forest plot (Figure 5), along with the pooled estimate and heterogeneity indices. The heterogeneity test among studies presented in Figure 6 showed an I^2^ value of 54%, which is statistically significant with *p* = 0.02. The assessment for publication bias using the funnel plot indicates symmetry (Figure 6), supporting the conclusion that there is no significant impact from the selection of the included studies on the results.

#### 3.1.2. Dataset 2: Multifocal SI-GISTs in the Number of GISTs with NF-1 Cases

A total of eight studies were included in the analysis, comprising 42 recorded cases [52,53,54,55,56,61,62,63]. The analysis for the pooled estimate of the proportion of multifocal cases among GISTs and NF-1 per study was based on random-effects models and the inverse variance method after logarithmic transformation. The pooled proportion estimate is 0.54 with a 95% confidence interval (0.31–0.76). The values for each study are presented in Table 4 and are graphically represented in the forest plot (Figure 7), along with the pooled estimate and heterogeneity indices. The heterogeneity test among studies presented in Figure 8 showed an I^2^ value of 35%, which is not statistically significant with *p* = 0.15. The assessment for publication bias using the funnel plot indicates symmetry (Figure 8), supporting the conclusion that there is no significant impact from the selection of the included studies on the results.

#### 3.1.3. Dataset 3: Simple/Solitary SI-GISTs in the Number of GISTs with NF-1 Cases

A total of eight studies were included in the analysis, comprising 42 recorded cases [52,53,54,55,56,61,62,63]. The analysis for the pooled estimate of the proportion of simple/solitary GIST cases among GIST + NF-1 per study was based on random-effects models and the inverse variance method after logarithmic transformation. The pooled proportion estimate is 0.41 with a 95% confidence interval (0.24–0.60). The values for each study are presented in Table 5 below and are graphically represented in the forest plot (Figure 9), along with the pooled estimate and heterogeneity indices. The heterogeneity test among studies presented in Figure 10 showed an I^2^ value of 18%, which is not statistically significant with *p* = 0.29. Testing for publication bias using the funnel plot indicates symmetry (Figure 10), which supports the conclusion that there is no major effect of the selection of specific studies on the results.

#### 3.1.4. Dataset 4: Simple/Solitary SI-GISTs in the Total Number of Patients with NF-1

A total of seven studies were included in the analysis, comprising 1032 recorded cases [53,54,55,56,61,62,63]. The analysis for the pooled estimate of the proportion of simple/solitary GIST cases among NF-1 patients per study was based on random-effects models and the inverse variance method after logarithmic transformation. The pooled proportion estimate is 0.02 with a 95% confidence interval (0.01–0.04). The values for each study are presented in Table 6 and are graphically represented in the forest plot (Figure 11), along with the pooled estimate and heterogeneity indices. The heterogeneity test among studies presented in Figure 12 showed an I^2^ value of 49%, which is not statistically significant with *p* = 0.07. Testing for publication bias using the funnel plot indicates symmetry (Figure 12), which supports the conclusion that there is no major effect of the selection of specific studies on the results.

#### 3.1.5. Dataset 5: Sporadic SI-GISTs in the General Population

A total of 15 studies [64,65,66,69,72,73,75,77,78,79,80,81,82,83,89,93] were included in the analysis, comprising 168,750,000 recorded cases. In the following table and graphs, the values are expressed in thousands of recorded cases. The analysis for the pooled estimate of the proportion of small intestinal GIST cases in the general population per study was based on random-effects models and the inverse variance method after logarithmic transformation. The pooled proportion estimate is 0.00002 with a 95% confidence interval (0.00001–0.00004). The values for each study are presented in the Table 7 below and are graphically represented in the forest plot (Figure 13), along with the pooled estimate and heterogeneity indices. The heterogeneity test among studies presented in Figure 14 showed an I^2^ value of 99%, which is statistically significant with *p* < 0.001. The assessment for publication bias using the funnel plot indicates a lack of symmetry (Figure 14), supporting the conclusion that there is an impact from the selection of the included studies on the results. However, even in the very large studies, the prevalence of small intestinal GISTs remains very low.

## 4. Discussion

The multifocal occurrence of GISTs in NF-1 patients presents a significant challenge for clinicians [32,33]. These tumors are primarily located in the small intestine, making endoscopic intervention difficult [6,7,30,31,32]. A common complication of these tumors is bleeding [36,37,38,39,40,41] and while embolization may achieve hemostasis, the anatomical constraints of small-bowel vasculature pose unique risks: distal occlusions carry elevated necrosis potential, whereas proximal embolization risks both ischemic injury and rebleeding via collateral circulation [94]. Identifying the exact bleeding source is challenging due to the high motility of the small intestine, which can promote the contrast agent distally, leading to mislocalization [44,95]. Imaging techniques often yield false-positive results [44,95]. Additionally, embolization of small intestinal branches can be difficult due to catheterization challenges, and furthermore, in many cases, these tumors are located in the distal segments of the small intestine [96,97]. Definitive diagnosis of the bleeding site can only be achieved through intraoperative endoscopy [44,95,98].

Despite endoscopic success, these tumors remain resistant to treatment with monoclonal agents such as imatinib [99,100]. Thus, conservative management aiming at tumor size reduction is highly challenging, and surgical resection is often the final solution [16,17]. Since NF-1 patients have a predisposition to developing GISTs in different segments of the small intestine over time, surgical resections should be limited. This is because, similar to Crohn’s disease, extensive resections can rarely lead to short bowel syndrome [100,101,102]. Thus, of particular urgency are cases presenting with acute intestinal obstruction or traction-induced intussusceptions; clinical scenarios that demand immediate intervention yet risk precipitating short bowel syndrome when multifocal lesions preclude radical resection. The work of Calomino et al. poignantly illustrates this dilemma: their two NF1-associated emergency resections (for ischemic presentations mimicking infarct) resulted in ultrashort bowel remnants (40–50 cm) with sacrificed ileocecal valves, yet through staged nutritional rehabilitation, transitioning from TPN to enteral to oral intake, both patients achieved long-term nutritional independence with preserved quality of life. These emergent presentations, albeit sporadically encountered, pose unique therapeutic dilemmas: obstruction from bulky tumors may necessitate expedited surgery despite the specter of recurrent disease, while intussusceptions mandate careful consideration of resection margins to avoid compromising bowel length [103]. Special attention should be given to tumor size, as it is the only macroscopic criterion for neoplastic progression in NF-1 patients with small intestinal involvement [104].

The prevalence of NF-1 in the general population is 1 in 3000–3500 individuals [9,10]. Some researchers suggest that 4–25% of NF-1 patients will develop a GIST at some point in their lives [31]. In a 1963 study from Ohio State University Hospital [30], patients were examined from two distinct periods: (i) clinical cases from 1949 to 1960, where 4% (1/29) of NF-1 patients had clinically evident GISTs, and(ii) autopsy cases from 1937 to 1960, where 25% (3/12) of generalized neurofibromatosis cases showed GISTs—establishing the aforementioned 4–25% range. However, this percentage is often reported to be much lower in the literature. Miettinen et al. (2006) emphasize that the 25% estimate is extremely high, as it is based on an older autopsy study that overestimates the GIST prevalence in NF-1 patients [7,30]. On the other hand, since the mean age of GIST onset in NF-1 patients is over 50 years, and NF-1 is now diagnosed at a young age, a large proportion of NF1-associated GIST cases remain unrecorded [53]. Thus, the prevalence of GISTs in NF-1 patients is often underestimated in modern studies [53]. Dupuis et al. (2023) highlight this issue in a cohort study, explaining that the prevalence of GISTs in NF-1 patients in their study is 3.7% [53], attributing the underestimation to the mean age of their study population (37.5 years) [53]. Most of the studies included in our meta-analysis have a mean patient age below 40 years. It is well known that the most common site of GISTs in NF-1 patients is the small intestine [6,7,30,31,32,33]. Some researchers suggest that almost all NF-1 patients with GISTs will eventually develop a GIST in the small intestine. Specifically, Mussi (2008) reports a 75% rate [32], while Salvi (2013) estimates 90% overall [33]. Our systematic review focuses exclusively on the small intestine. It analyzed 1386 recorded NF-1 cases and found that the prevalence of small intestine GISTs in NF-1 patients is 3%, with a confidence interval of 2–5%. Among these, 54% were multifocal, 41% were solitary small intestine GISTs, and the remainders were solitary duodenal GISTs. The prevalence of solitary small intestine and duodenal GISTs combined in the total NF-1 cases was approximately 2% (1–4%). This observation is supported by multiple researchers who describe that half of NF-1 patients with GISTs have multifocal small intestine GISTs, while the other half have solitary tumors [54,55]. However, these prevalence estimates provide only a fragmented picture of NF-1 patients’ lifetime risk. Over time, all GIST cases in NF-1 patients tend to become multifocal, as the likelihood of developing additional GISTs in other small intestinal segments is very high, as we will discuss further [32,33]. On the other hand, the prevalence of sporadic GISTs in the general population is approximately 12 per 100,000 individuals [64]. About one-third (30%) of these sporadic GIST cases occur in the small intestine, with the most common site being the stomach [105]. Chan et al. (2006) report an even lower proportion of sporadic small intestinal GISTs, estimating it at around 17% [69]. Among all sporadic GIST cases, less than 5% are associated with NF-1 mutations [106]. It is noteworthy that NF-1 patients are often excluded from general sporadic GIST registries, as their GIST cases are frequently multifocal and suggestive of rare syndromes [35]. One such syndrome is intestinal neurofibromatosis, a subtype of familial GISTs associated with PGFRA mutations [35]. These patients often lack external neurofibromatosis signs, and their disease first manifests with gastrointestinal symptoms [35]. Furthermore, many sporadic GIST cases that undergo genetic testing do not exhibit KIT gene mutations, even though many are CD117(+) [64]. When a GIST is quadruple-negative (negative for KIT, PDGFRA, BRAF, and SDH), there is a high likelihood of undiagnosed NF-1 [107]. It has been proposed that NF-1-associated GISTs should be identified not by genetic testing, but by monoclonal antibody assays that assess neurofibromin activity, as some mutations contribute to neurofibromin dysfunction without being genetically detectable [108]. Lastly, some GIST cases remain asymptomatic and go undiagnosed, despite NF-1 diagnosis being much easier today, primarily occurring in childhood [13,14,35]. As a result, many researchers report a much lower NF-1-related GIST prevalence, emphasizing that less than 1% of all recorded sporadic GIST cases are actually NF-1-associated [6,7]. In our meta-analysis, which focuses specifically on the small intestine, the prevalence of sporadic small intestine GISTs in a total of 168,750,000 recorded cases was 2 per 100,000 (1–4 per 100,000). Although the studies differ and a subgroup meta-analysis could not be performed due to the low number of NF-1-associated GIST cases, it is evident that the prevalence of small intestinal GISTs in NF-1 patients is significantly higher than that of sporadic small intestinal GISTs. This is apparent even for solitary small intestinal GISTs and duodenal GISTs, which share the same clinical and potentially pathogenic pathway as sporadic small intestinal GISTs (2% vs. 2 per 100,000). Nishida et al. (2016) report that the incidence of GISTs in NF-1 patients is 200 times higher than sporadic GIST [54]. If research focuses exclusively on the small intestine, this ratio may be much higher (potentially three times greater), since nearly all GISTs in NF-1 patients occur in the small intestine, whereas only one-third of sporadic small intestinal GIST cases do [64]. It is also important to emphasize that the incidence of NF-1-associated GISTs is very different and cannot be accurately determined, especially in comparison with sporadic GISTs. This is due to the different life expectancy of each patient group, the underreporting of many cases due to diagnostic challenges, and the wide time span covered by patient registries.

It is well known that neurofibromatosis is a disease caused by a mutation in the NF-1 gene, located on chromosome 17q11.2, following an autosomal dominant inheritance pattern [13,14]. The GISTs in NF-1 patients are CD117/DOG1 (+), despite the absence of KIT/PGFRA mutations (Figure 15) [2,3].

These tumors belong to the wild-type (15%) GIST category and result from germline mutations, specifically in the NF-1 gene mentioned earlier [8]. Maertens (2006) precisely described the molecular pathogenesis of GISTs in NF-1 patients and explained why these tumors occur at a much higher frequency compared to the general population, as well as why they are often multifocal [109]. Neurofibromin functions as a negative regulator of the RAS/MAPK pathway by catalyzing the hydrolysis of active Ras-GTP into inactive Ras-GDP. Active RAS binds to BRAF or the proto-oncogene RAF-1 at a specific Ras-binding domain (RBD) in BRAF, which is a serine/threonine kinase. This leads to RAF activation through homo- and heterodimerization [110]. Activated RAF subsequently triggers the MAPK pathway, leading to growth signaling activation through KIT/PGFRA mRNA expression. This mechanism explains why GISTs in NF-1 patients are CD117 (+), albeit indirectly. However, in this case, KIT exhibits extremely low levels of systemic autophosphorylation (constitutive autophosphorylation). Neurofibromin does not actively participate in somatic mutations. Additionally, the JAK-STAT3 and PI3K-AKT pathways are less active in NF-1-associated GIST cases. Despite these major molecular differences, the MAPK pathway remains common to both sporadic GISTs and NF-1-associated GISTs [109]. All conditions resulting from RAS/MAPK pathway dysregulation are classified as RASopathies. RASopathies are genetic syndromes caused by germline mutations in genes encoding components of the RAS/MAPK signaling pathway [110]. Besides NF-1, other RASopathies include Noonan syndrome, Costello syndrome, and Legius syndrome. A wide variety of tumors can develop in these syndromes, including neuroblastoma, acute lymphoblastic leukemia (ALL), non-Hodgkin’s lymphoma (NHL), acute myeloid leukemia (AML), non-small-cell lung cancer, tubulovillous adenomas of the colon, breast cancer, ovarian dermoid tumors, and even melanoma. It is well established that multiple distinct genes contribute to the development of these tumors. Recently, the role of the BRAF gene in BRAF-associated RASopathies has been extensively described [110].

The mechanism by which some GISTs develop as solitary tumors while others become multifocal has not yet been fully elucidated. In our meta-analysis, we found that 41% of cases presented as solitary GISTs, while 54% were multifocal GISTs. Additionally, NF-1 patients may develop multiple solitary GISTs, making the distinction between the two categories unclear [54,55]. Chen et al. (2002) studied the nature of these multifocal tumors, investigating whether their origin is polyclonal or monoclonal, which would indicate hyperplasia or neoplasia, respectively [111]. The study concluded that familial and multiple GISTs result from hyperplasia of interstitial cells of Cajal, categorizing them as non-neoplastic diffuse ICC hyperplasia. However, tumors larger than 5 mm appeared to have a monoclonal nature, suggesting neoplastic transformation. An intermediate stage between hyperplasia and neoplasia is the formation of micro-GISTs, measuring 1–10 mm [112]. For this reason, apart from mitotic count and markers such as Ki67, tumor size is the only macroscopic criterion suggestive of neoplasia, which has been incorporated into GIST staging systems [104]. The biological continuum from polyclonal hyperplasia to monoclonal neoplasia mandates a multidisciplinary approach—integrating serial endoscopic surveillance for subcentimeter lesions, Ki67-indexed risk stratification, and judicious surgical intervention—to optimize the critical transition from oncologic vigilance to therapeutic action while preserving intestinal function. Surgeons must keep this in mind, as extensive and multiple resections in the small intestine may ultimately cause significant morbidity in NF-1 patients without necessarily increasing oncologic risk. Recently, the role of selumetinib and mirdametinib in the treatment of plexiform neurofibromas has been recognized, primarily for clinical symptom improvement [113,114]. However, their role in GIST downstaging in NF-1 patients has not yet been investigated. As a result, surgical resection remains the only available treatment for NF-1-associated GISTs, in a patient population already undergoing multiple surgical procedures [16,17].

The most common site of GISTs in NF-1 patients is the small intestine, whereas in sporadic GISTs, only one-third of cases present as small intestinal GISTs [32,33,64]. This remains a scientific question related to the GIST cell of origin [112]. It is widely known that interstitial cells of Cajal are responsible for GIST development [1]. However, even these cells have different subtypes with varying distributions depending on the intestinal segment and layer in which they are located [115]. Additionally, they exhibit differential expression of the transcription factor ETV1 (E twenty-six), which is a lineage transcription factor that plays a crucial role in GIST formation [116]. There are myenteric (ICC-MYs) and intramuscular (ICC-IMs) ICCs, as well as submucosal ICCs (ICC-SMPs) and deep myenteric plexus ICCs (ICC-DMPs) [1]. In the colon, ICC-MYs and ICC-IMs stain positive for ETV1, but ICC-SMPs do not [116]. In the small intestine, ICC-MYs stain positive for ETV1, but ICC-DMPs do not [116]. In experimental models with KIT gene mutations, ICC-MYs and ICC-IMs undergo hyperplasia, whereas ICC-SMPs and ICC-DMPs do not [116]. Another recent study based on animal models suggests that the progenitor cells of GISTs are smooth muscle cell progenitors, which undergo genetic deviation, leading to GIST formation [117]. This has introduced the alternative cell of origin theory for GISTs [117]. Such progenitor cells are primarily ICC-DMPs and ICC-SMPs, which eventually develop ICC hyperplasia following differentiation influenced by LRIF1 (Ligand-dependent nuclear Receptor-Interacting Factor 1) and BRAF V600E [117]. It is well known that ICC-DMPs are widely distributed in the small intestine, particularly in the jejunum and ileum [118]. Moreover, studies in genetically modified W/Wv mice with c-KIT mutations (ICC-deficient model mice) have shown that the only preserved myenteric plexus detectable microscopically was the ICC-DPMs in the ileum, while in contrast, ICC-IMs in the gastric antrum and colon were absent [119]. However, for ICC hyperplasia to progress into oncogenesis, multiple genetic mechanisms must be involved. Ran et al. (2017) successfully induced both ICC hyperplasia and multifocal GIST formation in the gastrointestinal tract of mice by activating the BRAF V600E gene and simultaneously deleting Trp53 (tryptophan 53) [116]. The activation of BRAF V600E in ETV1(+) cells resulted only in ICC hyperplasia, without malignant transformation. However, the additional loss of the tumor suppressor gene Trp53 led to progression from hyperplasia to malignancy, resulting in the formation of multiple GISTs throughout the gastrointestinal tract [116]. Notably, the experimental model achieved 100% success in mice [116]. Thus, not only does the cell of origin influence GIST formation, but a specific genetic combination is also required for malignant transformation. This explains why the phenotype of GISTs in multifocal syndromes cannot be fully elucidated. Some tumors are clearly GISTs with CD117(+) expression and do not exhibit the CD117(−), S100(−), SMA(−), or desmin(−) phenotype characteristic of mesenchymal tumors derived from smooth muscle cells [120].

The integration of prognostic scoring systems has become indispensable for stratifying survival outcomes and metastatic risk in GIST management, with particular relevance for NF1-associated cases. The recently developed RECKGIST score represents a transformative advance in this domain, offering NF1-specific risk stratification that surpasses conventional classification systems in predictive accuracy. As demonstrated by Cuvelier et al., this tool effectively discriminates low-risk from high-risk cohorts based on critical histopathological parameters, enabling more nuanced clinical decision-making. Notably, the score’s robust differentiation of recurrence risk underscores its potential to refine surveillance protocols and therapeutic interventions for NF1 patients. While further validation is warranted, RECKGIST emerges as a paradigm-shifting framework that addresses longstanding gaps in the prognostication of NF1-GIST behavior [121].

This meta-analysis has several limitations. Firstly, it has a moderate risk of bias, primarily due to the limited number of patients included in each study and the potential selection bias in individual studies. The novelty of this study lies in its exclusive focus on small intestinal GISTs and the comparison between NF-1 patients and the general population. However, due to the fact that the included studies were heterogeneous, no subgroup meta-analysis with specific statistical tools was performed; instead, we relied on direct percentage comparisons, which clearly showed a significant difference between the two groups. The focus on the small intestine, while strength of the study, is also a limitation, as GIST occurrence in other gastrointestinal sites was not extensively analyzed. This decision was made to avoid overcomplicating the review, taking into consideration that this aspect has already been extensively studied in the past. The primary objective of this study was to emphasize the clinical challenges associated with small intestinal GISTs, particularly endoscopic accessibility. Additionally, this systematic review aims to provide indirect insights into the molecular pathogenesis of GISTs, proposing a potential explanation for the origin of GIST cells, which may be predominantly located in the small intestine. A key issue in this research area is the molecular mechanisms driving multifocal tumors in NF-1 mutations, which is a turning point in the broader study of GIST pathogenesis. Although significant progress has been made in this field, research has primarily focused on the increased prevalence of GISTs in NF-1 patients, rather than differentiating between multifocal/hyperplasic and solitary/neoplastic GISTs. Furthermore, the transition from hyperplasia to neoplasia requires further investigation through new studies. Most research on this topic remains limited to animal models. Future studies are expected to address these unanswered questions, which will have a direct impact on the clinical management of these patients. More specifically, the development of molecular biomarkers to preoperatively distinguish multifocal hyperplasia from solitary neoplasia could revolutionize surgical planning in NF-1 patients. Precision surveillance protocols must be established to detect the earliest transition from Cajal cell hyperplasia to clonal neoplasia, particularly in high-risk intestinal segments. Last but not least, the creation of evidence-based decision algorithms integrating tumor focality, mutational status, and anatomical constraints will be paramount for optimizing both oncological outcomes and bowel function preservation. These translational imperatives—now crystallized by our systematic evidence synthesis—chart an actionable roadmap for transforming NF-1-GIST management from reactive intervention to proactive precision medicine.

## 5. Conclusions

Following this elaborative analysis, it emerges that small intestine gastrointestinal stromal tumors (SI-GISTs) demonstrate unique characteristics in neurofibromatosis type 1 (NF-1) patients that merit careful consideration. The prevalence of SI-GISTs in NF-1 patients is established at 3% (range 2–5%), presenting a clear epidemiological benchmark. Among these cases, a distinct pattern emerges: 54% manifest as multifocal SI-GISTs and 41% as solitary SI-GISTs, with the remaining cases representing solitary duodenal GISTs. When examining solitary SI-GISTs specifically, the prevalence in NF-1 patients approximates 2% (1–4%), providing further granularity to our understanding. These figures gain particular significance when contrasted with the general population’s sporadic small intestinal GIST prevalence of merely 2 per 100,000 individuals, revealing that NF-1 patients face a substantially elevated risk (3% versus 0.002%).

This systematic review, with its exclusive focus on small intestinal manifestations, compellingly demonstrates that the propensity for SI-GIST development in NF-1 patients far exceeds prior estimations compared to the general population. The notably high frequency of SI-GISTs in NF-1 patients, coupled with their multifocal nature, offers invaluable insights that may illuminate both the cellular origins of GISTs and the fundamental pathogenic mechanisms governing GIST tumorigenesis. From a clinical perspective, these findings carry immediate practical implications: when evaluating NF-1 patients presenting with gastrointestinal bleeding, physicians must maintain heightened suspicion for underlying small intestinal tumors, as these cases presents unique challenges in diagnosis and management.

The management challenges are multifaceted, encompassing difficulties in precise bleeding site localization, constraints on endoscopic intervention options, limited efficacy of tyrosine kinase inhibitors (KIT inhibitors) in tumor downstaging, and complex intraoperative decision-making regarding resection of potential neoplastic foci. Together, these groundbreaking revelations not only revolutionize our understanding of GIST pathogenesis but sound an urgent clarion call for the immediate implementation of specialized diagnostic algorithms, tailored therapeutic strategies, and rigorous surveillance protocols to safeguard NF-1 patients from this insidious and disproportionately prevalent malignancy.

## Figures and Tables

**Figure 1 cancers-17-01934-f001:**
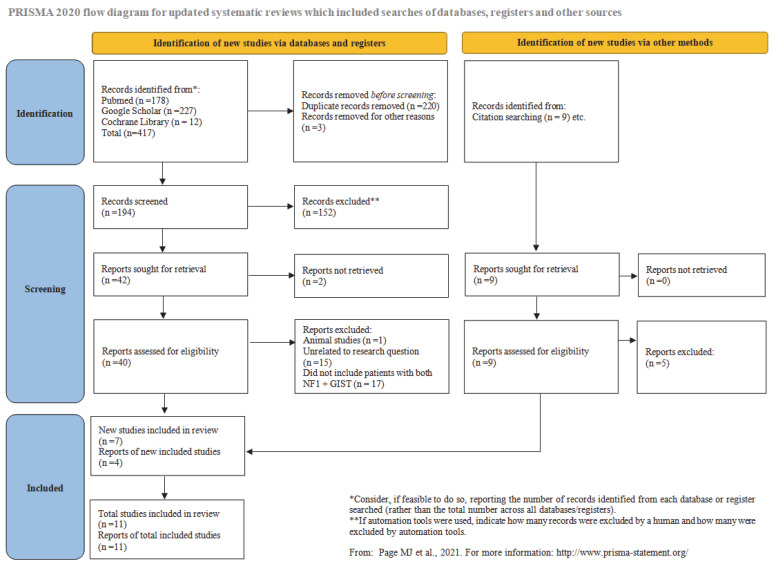
PRISMA flow diagram [48].

**Figure 2 cancers-17-01934-f002:**
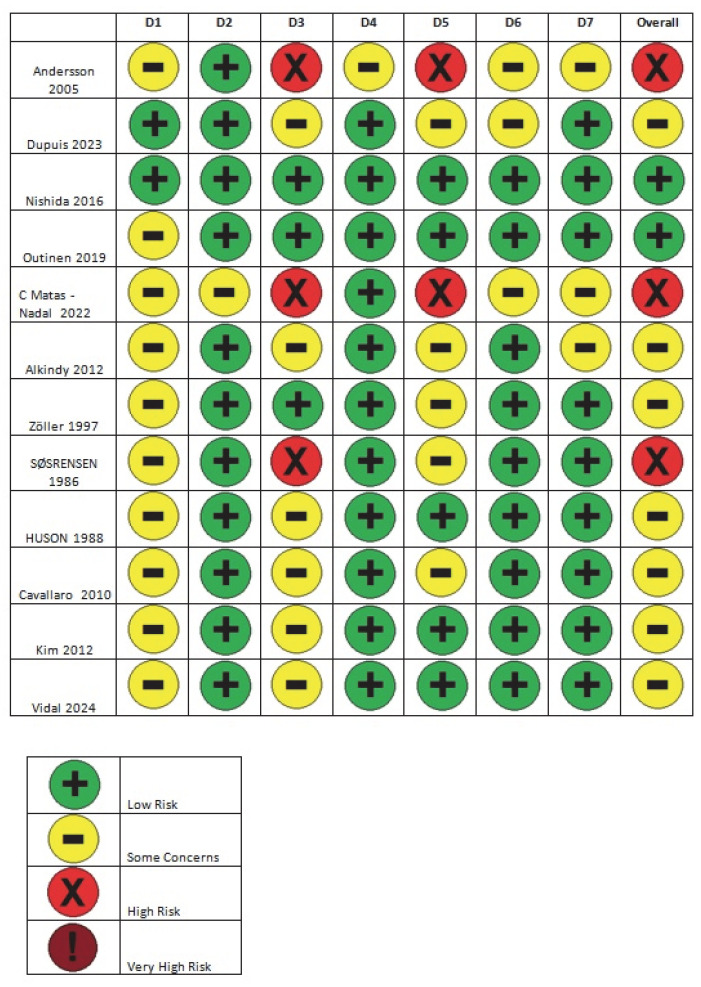
ROBINS-E bias evaluation panel [52,53,54,55,56,57,58,59,60,61,62,63].

**Figure 3 cancers-17-01934-f003:**
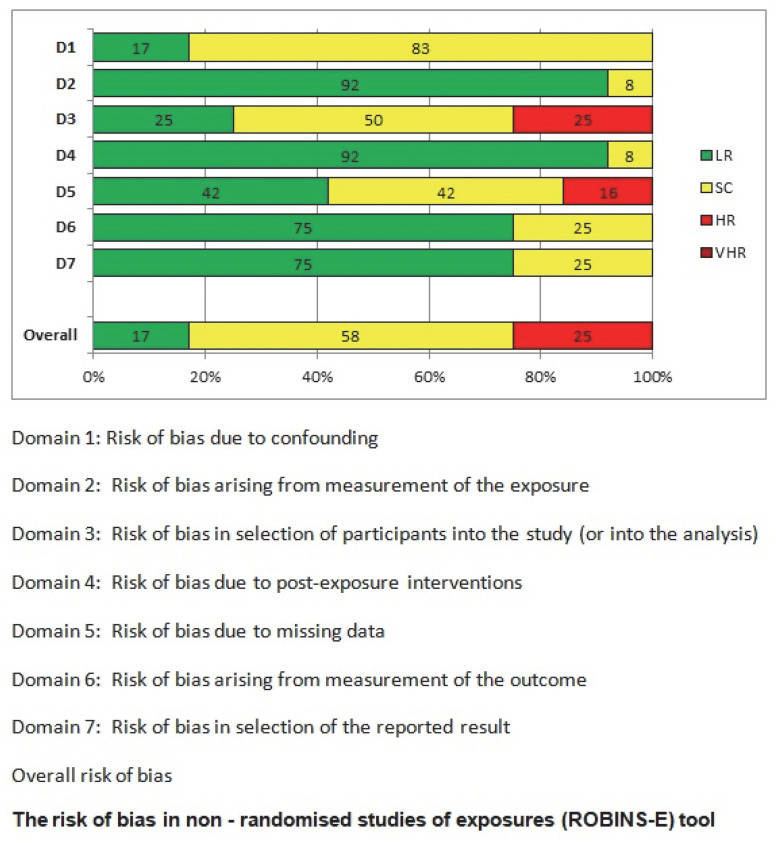
Percentage analysis of the bias-related questions’ assessments.

**Figure 4 cancers-17-01934-f004:**
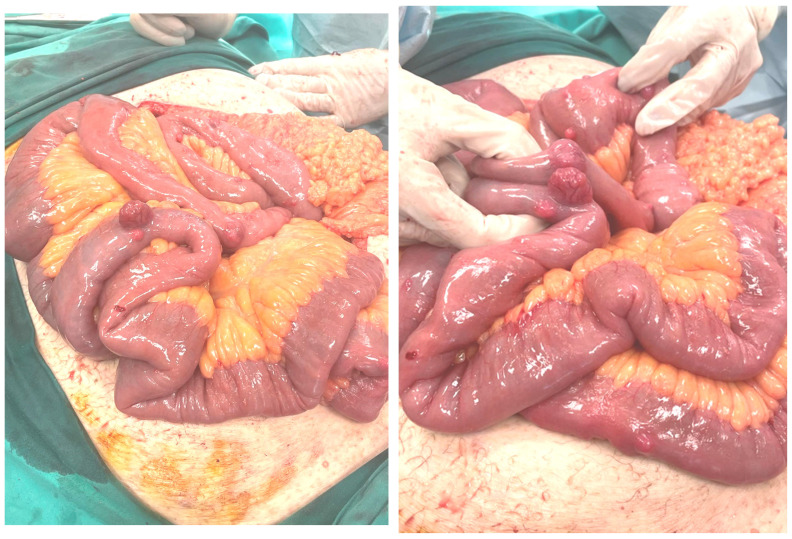
Multifocal small intestinal GIST resected during emergency surgery for acute intraluminal bleeding in an NF-1 patient (3rd Surgical Department, AHEPA University Hospital, School of Medicine, Aristotle University of Thessaloniki archive).

**Figure 5 cancers-17-01934-f005:**
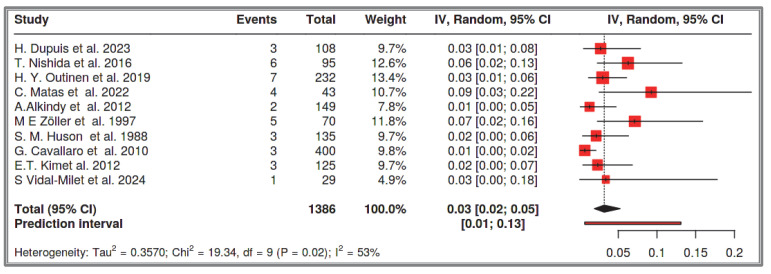
Dataset 1 forest plot [53,54,55,56,57,58,60,61,62,63].

**Figure 6 cancers-17-01934-f006:**
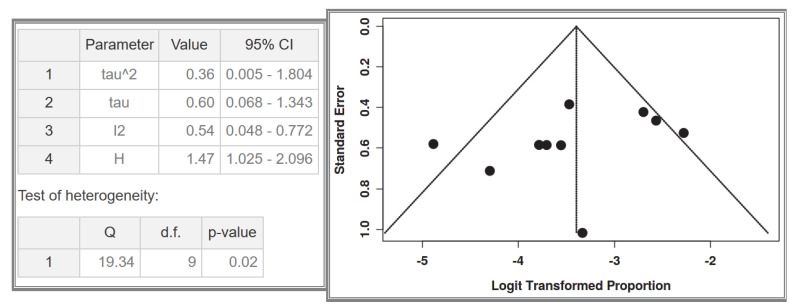
Heterogeneity test and funnel plot interrogating dataset 1.

**Figure 7 cancers-17-01934-f007:**
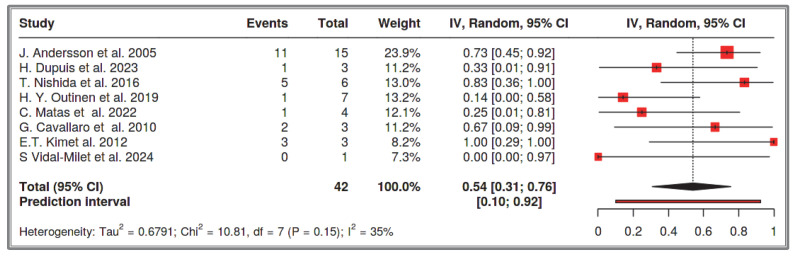
Dataset 2 forest plot [52,53,54,55,56,61,62,63].

**Figure 8 cancers-17-01934-f008:**
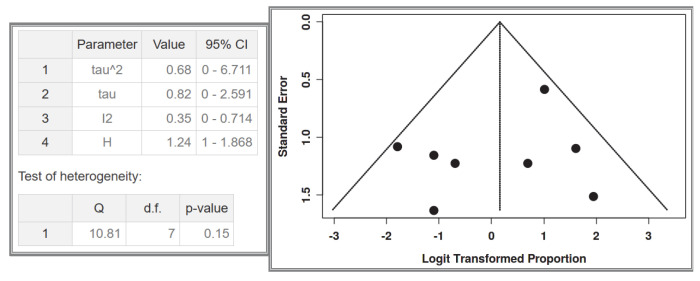
Heterogeneity test and funnel plot interrogating dataset 2.

**Figure 9 cancers-17-01934-f009:**
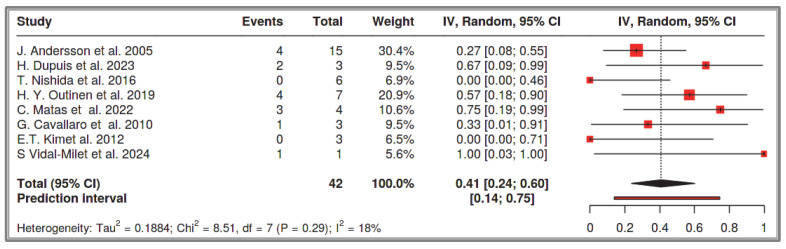
Dataset 3 forest plot [52,53,54,55,56,61,62,63].

**Figure 10 cancers-17-01934-f010:**
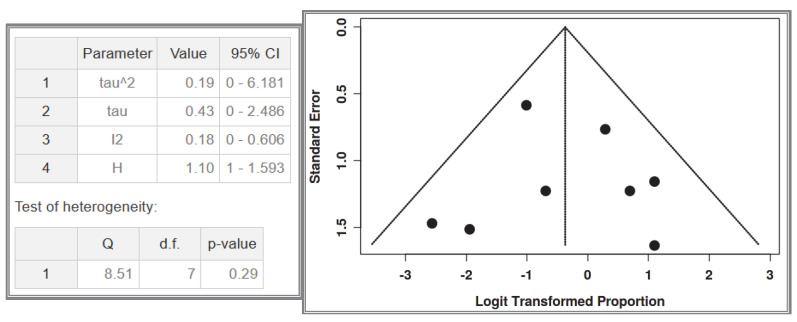
Heterogeneity test and funnel plot interrogating dataset 3.

**Figure 11 cancers-17-01934-f011:**
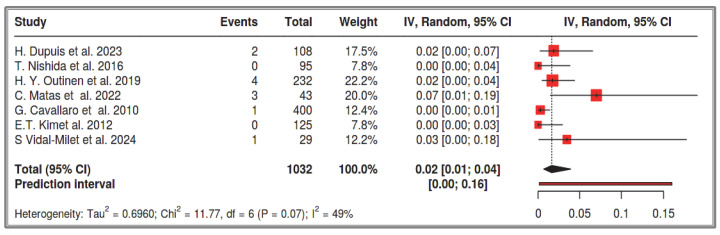
Dataset 4 forest plot [53,54,55,56,61,62,63].

**Figure 12 cancers-17-01934-f012:**
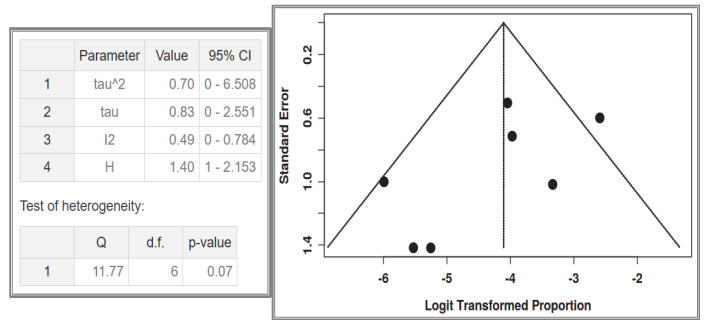
Heterogeneity test and funnel plot interrogating dataset 4.

**Figure 13 cancers-17-01934-f013:**
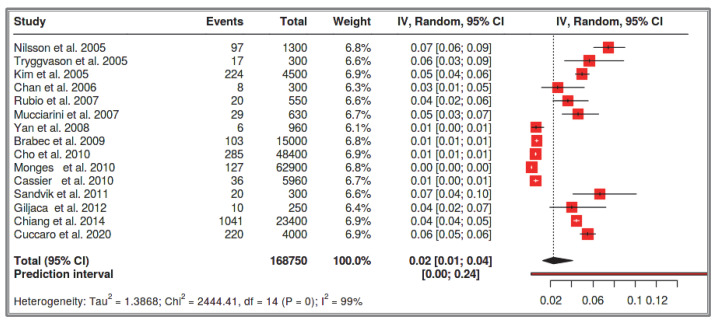
Dataset 5 forest plot [64,65,66,69,72,73,75,77,78,79,80,81,82,83,89].

**Figure 14 cancers-17-01934-f014:**
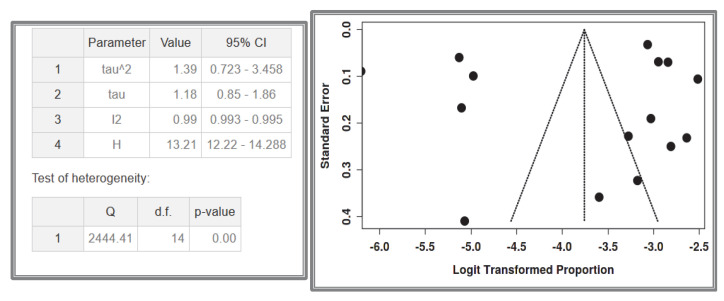
Heterogeneity test and funnel plot interrogating dataset 5.

**Figure 15 cancers-17-01934-f015:**
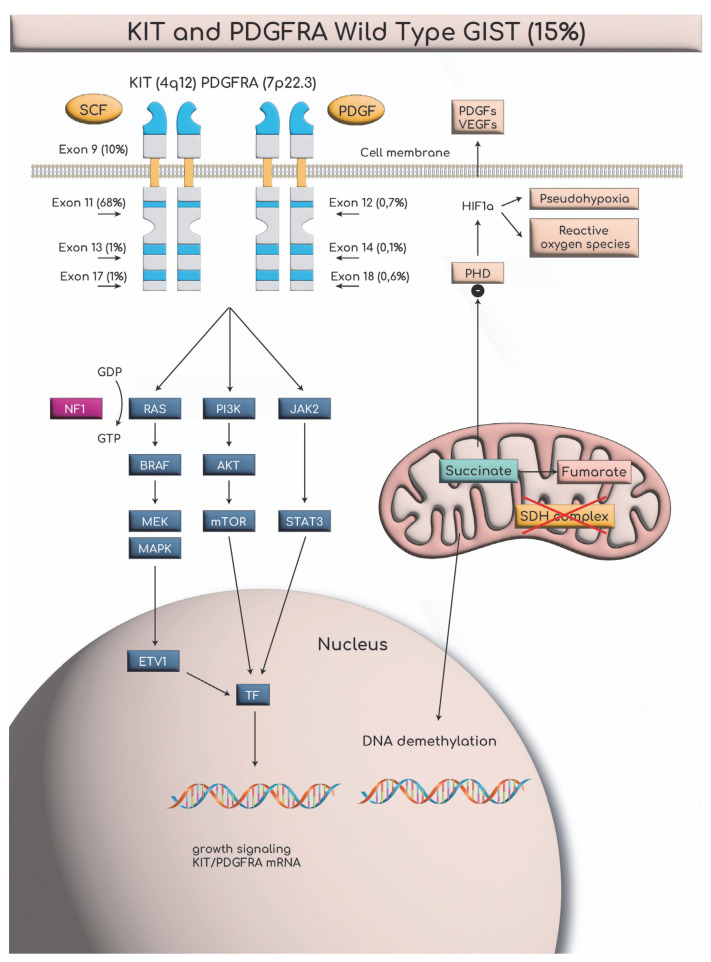
Beyond KIT/PDGFRA: the distinct molecular signature of the NF-1 tumor microenvironment in GIST pathobiology (SCF; Stem Cell Factor, VEGF; Vascular Endothelial Growth Factor, HIF1a; Hypoxia Inducible Factor 1a, PHD; Prolyl Hydroxylase Domain, GDP/GTP; Guanosine Diphosphate/Triphosphate, ETV1; ETS Variant transcription factor 1, MTOR; mammalian Target Of Rapamycin, TF; Transcription Factor).

**Table 1 cancers-17-01934-t001:** Spreadsheet generated by the first researcher.

Author	Title	Study Type	Year	Patients with NF	GIST + NF	Multifocal SI-GIST	Simple SI-GIST	Simple Duodenal GIST
**Andersson**	NF-1 associated gastrointestinal stromal tumors have unique clinical, phenotypic and genotypic characteristics [52]	Observational retrospective	2005		15	11	4	
**Dupuis**	Prevalence of endocrine manifestations and GIST in 108 systematically screened patients with NF-1 [53]	Monocentric retrospective	2023	108	3	1	2	0
**Nishida**	Gastrointestinal stromal tumors in Japanese patients with NF-1 [54]	Retrospective	2016	95	6	5	1	0
**Outinen**	Intestinal tumors in neurofibromatosis 1 with special reference to fatal gastrointestinal tumors (GIST) [55]	Retrospective register-biased total population	2019	232	7	1	4	2
**C Matas-** **Nadal**	Abdominal tumors in patients with neurofibromatosis type 1 (NF-1): genotype-phenotype relationships [56]	Retrospective	2022	43	4	1	3	0
**Alkindy**	Genotype-phenotype associations in NF-1: an increased risk of tumor complications in patients with NF-1 splice-site mutations [57]	Retrospective	2006	149	2	Unknown	Unknown	Unknown
**Zoller**	Malignant and benign tumors in patients with neurofibromatosis type 1 in a defined Swedish population [58]	Retrospective	1997	70	5	Unknown	Unknown	Unknown
**Sosrensen **	On the natural history of Von Recklinghausen neurofibromatosis [59]	Retrospective	1986	212	0	Unknown	Unknown	Unknown
**Huson**	Von Recklinghausen neurofibromatosis: a clinical and population study in south-east Wales [60]	Retrospective	1988	135	3	Unknown	Unknown	Unknown
**Cavallaro**	Surgical management of abdominal manifestations of NF-1: experience of a single center [61]	Retrospective	2010	400	3	2	1	0
**Kim**	Oncologic manifestations of neurofibromatosis type 1 in Korea [62]	Retrospective	2012	125	3	3	0	0
**Vidal-Milan**	Case report: benign and malignant tumors in adult patients with neurofibromatosis type 1: a comprehensive case series from a large oncologic reference center [63]	Retrospective	2024	29	1	0	1	0

**Table 2 cancers-17-01934-t002:** Spreadsheet generated by the second researcher; *n:* number, N/R: not reported.

Author (Publication Year)	Study Type	Study (Country)	Population (Million)	Time Span	SI-GIST Patients (n%)	Annual Crude Incidence of All GISTs (GIST Patients/Year)	Annual Crude Incidence of SI-GISTs (SI-GIST Patients/Year)
**Nilsson et al. (2005)**	Retrospective [64]	Sweden	1.3–1.6	1983–2000	97 (33.7%)	14.5	4.9
**Tryggvason et al. (2005)**	Retrospective [65]	Iceland	0.3	1990–2003	17 (29.8%)	11	3.3
**Kim et al. (2005)**	Retrospective [66]	Korea	4.5	2001–2022	224 (30%)	7.7	2.3
**Tran et al. (2005)**	Retrospective [67]	USA	N/R	1992–2000	525 (36%)	6.8	2.4
**Perez et al. (2006)**	Retrospective [68]	USA	N/R	1992–2002	N/R (34%)	6.9	2.3
**Chan et al. (2006)**	Retrospective [69]	Hong Kong	0.3–0.35	1995–2003	8 (17%)	16.8–19.6	
**Steigen et al. (2007)**	Retrospective [70]	Norway	N/R	1974–2003	23 (25.8%)	19	4.9
**Tzen et al. (2006)**	Retrospective [71]	Taiwan	N/R	1998–2004	41 (38.1%)	13.7	5.2
**Rubio el al. (2007)**	Retrospective [72]	Spain	0.55	1994–2001	20 (43.5%)	10.9	4.7
**Mucciarini et al. (2007)**	Retrospective [73]	Italy	0.63	1991–2004	29 (23.4%)	14.2	3.3
**Ahmed et al. (2008)**	Retrospective [74]	UK	N/R	1987–2003	30 (16.2%)	13.2	2.1
**Yan et al. (2008)**	Retrospective [75]	Canada	0.96	2000–2004	6 (27%)	5.1–8.5	
**Mazzola et al. (2008)**	Retrospective [76]	Switzerland	N/R	1995–2002	11 (25.6%)	14.7	3.7
**Brabec et al. (2009)**	Retrospective [77]	Chechia & Slovakia	15	2000–2008	103 (37%)	5.2	1.9
**Cho et al. (2010)**	Retrospective [78]	Korea	48.4	2003–2004	285 (23.2%)	16–22	
**Monges et al. (2010)**	Prospective [79]	France	62.9	2005	127 (21.5%)	8.5–10	
**Cassier et al. (2010)**	Prospective [80]	France	5.96	2005–2006	36 (27%)	11.2	3
**Sandvik et al. (2011)**	Retrospective [81]	Norway	0.3	1980–2009	20 (38%)	6.5	2.5
**Giljaca et al. (2012)**	Retrospective [82]	Croatia	0.25	1997–2007	10 (32.2%)	12.4	4
**Chiang et al. (2014)**	Retrospective [83]	Taiwan	23.4	1998–2008	1041 (34.8%)	11.3–19.7	
**Ma et al. (2014)**	Retrospective [84]	USA	N/R	2001–2011	1781 (29%)	6.8	1.97
**Kukar et al. (2015)**	Retrospective [85]	USA	N/R	1990–2009	1463 (33%)	3.9–4.8	
**Verschoor et al. (2018)**	Retrospective [86]	Netherlands	N/R	2003–2012	518 (21.1%)	12.2–17.7	
**Patel et al. (2019)**	Retrospective [87]	USA	N/R	2001–2015	10158 (30%)	7	2.1
**Ulanja et al. (2019)**	Retrospective [88]	USA	N/R	2002–2015	1988 (27.6%)	7.6	2.1
**Cuccaro et al. (2020)**	Retrospective [89]	Italy	4	2006–2015	220 (31.1%)	17	5.3
**Milostic et al. (2021)**	Retrospective [90]	Croatia	N/R	2004–2017	20 (30.8%)	14.7	4.5

**Table 3 cancers-17-01934-t003:** Source studies contributing to dataset 1.

	Author	Year	*n*	Events
**1**	H. Dupuis et al. [53]	2023	108	3
**2**	T. Nishida et al. [54]	2016	95	6
**3**	H.Y. Outinen et al. [55]	2019	232	7
**4**	C. Matas et al. [56]	2022	43	4
**5**	A. Alkindy et al. [57]	2012	149	2
**6**	M. E. Zoller et al. [58]	1997	70	5
**7**	S.M. Huson et al. [60]	1988	135	3
**8**	G. Cavallaro et al. [61]	2010	400	3
**9**	E.T. Kim et al. [62]	2012	125	3
**10**	S Vidal-Milet et al. [63]	2024	29	1

**Table 4 cancers-17-01934-t004:** Source studies contributing to dataset 2.

	Author	Year	*n*	Events
**1**	J. Andersson et al. [52]	2005	15	11
**2**	H. Dupuis et al. [53]	2023	3	1
**3**	T. Nishida et al. [54]	2016	6	5
**4**	H. Y. Outinen et al. [55]	2019	7	1
**5**	C. Matas et al. [56]	2022	4	1
**6**	A. Alkindy et al. [57]	2012		
**7**	M.E. Zoller et al. [58]	1997		
**8**	S.M. Huson et al. [60]	1988		
**9**	G. Cavallaro et al. [61]	2010	3	2
**10**	E.T. Kim et al. [62]	2012	3	3
**11**	S. Vidal-Milet et al. [63]	2024	1	0

**Table 5 cancers-17-01934-t005:** Source studies contributing to dataset 3.

	Author	Year	*n*	Events
**1**	J. Andersson et al. [52]	2005	15	4
**2**	H. Dupuis et al. [53]	2023	3	2
**3**	T. Nishida et al. [54]	2016	6	0
**4**	H. Y. Outinen et al. [55]	2019	7	4
**5**	C. Matas et al. [56]	2022	4	3
**6**	A. Alkindy et al. [57]	2012		
**7**	M.E. Zoller et al. [58]	1997		
**8**	S.M. Huson et al. [60]	1988		
**9**	G. Cavallaro et al. [61]	2010	3	1
**10**	E.T. Kim et al. [62]	2012	3	0
**11**	S. Vidal-Milet et al. [63]	2024	1	1

**Table 6 cancers-17-01934-t006:** Source studies contributing to dataset 4.

	**Author**	**Year**	** *n* **	**Events**
**1**	H. Dupuis et al. [53]	2023	108	2
**2**	T. Nishida et al. [54]	2016	95	0
**3**	H. Y. Outinen et al. [55]	2019	232	4
**4**	C. Matas et al. [56]	2022	43	3
**5**	G. Cavallaro et al. [61]	2010	400	1
**6**	E.T. Kim et al. [62]	2012	125	0
**7**	S. Vidal-Milet et al. [63]	2024	29	1

**Table 7 cancers-17-01934-t007:** Source studies contributing to dataset 5.

	Author	Year	*n*	Events
**1**	Nilsson et al. [64]	2005	1300.00	97.00
**2**	Tryggvason et al. [65]	2005	300.00	17.00
**3**	Kim et al. [66]	2005	4500.00	224.00
**4**	Chan et al. [69]	2006	300.00	8.00
**5**	Rubio et al. [72]	2007	550.00	20.00
**6**	Mucchiarini et al. [73]	2007	630.00	29.00
**7**	Yan et al. [75]	2008	960.00	6.00
**8**	Brabec et al. [77]	2009	15,000.00	103.00
**9**	Cho et al. [78]	2010	48,400.00	285.00
**10**	Monges et al. [79]	2010	62,900.00	127.00
**11**	Cassier et al. [80]	2010	5960.00	36.00
**12**	Sandvik et al. [81]	2011	300.00	20.00
**13**	Giljaca et al. [82]	2012	250.00	10.00
**14**	Chiang et al. [83]	2014	23,400.00	1041.00
**15**	Cuccaro et al. [89]	2020	4000.00	220.00

## Data Availability

To facilitate research reproducibility, the aggregated data from this meta-analysis can be provided by the corresponding author following established data sharing protocols. Link to the PROSPERO international systematic review registry: https://www.crd.york.ac.uk/PROSPERO/view/CRD420251002384, accessed on 15 March 2025.

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
