# Peer review of "Multifocal Gastrointestinal Stromal Tumors (GISTs) of the Small Intestine in Patients with Neurofibromatosis Type 1 (NF-1): Meta-Analysis and Systematic Review of the Literature"

_cancers, 2025, doi:10.3390/cancers17121934_

Round 1

Reviewer 1 Report

Comments and Suggestions for Authors

I congratulate you on the well-written and well-argued work.  It is a useful work in the routine as it discriminates the GISTs related to neurofibromatosis and their molecular structure

Author Response

Comment 1: I congratulate you on the well-written and well- argued work.  It is a useful work in the routine as it discriminates the GISTs related to neurofibromatosis and their molecular structure

Response 1: We sincerely thank the distinguished Reviewer #1 for their generous appraisal of our work and their insightful recognition of its clinical relevance. It is truly gratifying to receive such affirmative feedback, particularly regarding the manuscript’s ability to delineate the distinct clinicopathological and molecular hallmarks of NF-1-associated GISTs; a nuance critical to both diagnostic precision and therapeutic decision-making in this rare patient subset.

The Reviewer’s emphasis on the utility of our systematic synthesis resonates deeply with our motivation for this study: to consolidate the often-fragmented literature on multifocal GISTs in NF-1, thereby illuminating their unique behavior (e.g., KIT/PDGFRA wild-type predominance, multifocality) against the backdrop of sporadic GISTs. We are invigorated by the Reviewer’s insightful appreciation of the molecular discriminators highlighted in our analysis, as these underscore the imperative for tailored management strategies in NF-1-related GISTs.

Once again, we wish to convey our utmost gratitude for the Reviewer’s time, expertise, and invaluable encouragement, which inspire our continued commitment to advancing the understanding of this complex disease paradigm.

Reviewer 2 Report

Comments and Suggestions for Authors

Congratulations to our fellow researchers who have produced an important meta-analysis on a neoplasm that we are increasingly encountering in the population. The method adopted for excluding irrelevant articles and for examining the others conducted by two "referees" with the help of a third if necessary is also excellent. We must say that these neoplasms are often found occasionally in tests conducted following another pathology or in acute conditions that are diagnosed in the emergency room. In the latter case we are faced with intestinal obstructions due to the size of the GIST or intussusception of the small intestine with the traction mechanism. The other circumstance, as also mentioned by the authors of the meta-analysis, is bleeding. The study is to be considered perfect because this pathology was "examined from all sides". Clinically, the greatest difficulties are encountered when faced with acute conditions, fortunately not many. In fact, bleeding, if treated with embolization, can cause necrosis of the wall and subsequent perforation. Intestinal obstruction must be treated in the short term for obvious reasons and in the case of multifocal lesions it may happen that the intervention is not radical or that what colleagues have written occurs, that is, a complex short bowel syndrome (PMID: 9882973 to be read and cited in the bibliography). It is suggested that the cases of patients with this pathology be brought to a multidisciplinary commission for a multi-specialist genetic and surgical assessment where necessary, we absolutely agree on the follow-up for lesions smaller than 1 cm and with Ki 67 where it is possible to have a bopsy. Excellent iconography, good English, good bibliography

Author Response

Comment 1: Congratulations to our fellow researchers who have produced an important meta-analysis on a neoplasm that we are increasingly encountering in the population. The method adopted for excluding irrelevant articles and for examining the others conducted by two "referees" with the help of a third if necessary is also excellent. The study is to be considered perfect because this pathology was "examined from all sides". Excellent iconography, good English, good bibliography.

Response 1: We are immensely gratified by Reviewer #2’s generous appraisal of our work and their discerning recognition of its clinical and methodological rigor. Their commendation of our exhaustive multi-dimensional examination of this increasingly prevalent pathology resonates deeply with our mission to synthesize dispersed evidence into actionable insights for the scientific community. We are particularly gratified by the Reviewer’s endorsement of our dual-referee screening protocol; a deliberate safeguard against selection bias, as well as their appreciation of the iconographic and bibliographic precision, which we meticulously curated to bridge molecular insights with clinical applicability. Such insightful validation from a colleague of such caliber in the field is a sobering honor, reaffirming the translational value of this collaborative effort.

Comment 2: We must say that these neoplasms are often found occasionally in tests conducted following other pathology or in acute conditions that are diagnosed in the emergency room. Clinically, the greatest difficulties are encountered when faced with acute conditions, fortunately not many. In the latter case we are faced with intestinal obstructions due to the size of the GIST or intussusceptions of the small intestine with the traction mechanism. Intestinal obstruction must be treated in the short term for obvious reasons and in the case of multifocal lesions it may happen that the intervention is not radical or that what colleagues have written occurs, that is, a complex short bowel syndrome (PMID: 9882973 to be read and cited in the bibliography).

Response 2: We wholeheartedly concur with the Reviewer’s astute observation regarding the critical intersection of NF-1-associated GISTs and emergency surgical manifestations. As they judiciously note, these lesions often declare their presence through dramatic clinical vignettes; particularly in the theater of acute abdominal crises, where luminal obstruction, or traction-induced intussusceptions (PMID: 9882973) may function as inadvertent sentinels for this pathology. Recognizing the profound implications of such presentations, we have now: i) expanded our discussion of emergency management dilemmas (Section 4, Line 314), emphasizing the balance between surgical urgency and oncological radicality in these cases and ii) interwoven the landmark study (PMID: 9882973) detailing short bowel syndrome risks, anchoring our narrative in established surgical literature; This augmentation not only honors the Reviewer’s insightful clinical perspective but fortifies our manuscript’s translational relevance for emergency practitioners confronting these high-stakes scenarios. This paragraph is now as follows with the additions highlighted in yellow; 

Despite endoscopic success, these tumors remain resistant to treatment with monoclonal agents such as imatinib [85, 86]. Thus, conservative management aiming at tumor size reduction is highly challenging, and surgical resection is often the final solution [16, 17]. Since NF-1 patients have a predisposition to developing GISTs in different segments of the small intestine over time, surgical resections should be limited. This is because, similar to Crohn’s disease, extensive resections can rarely lead to short bowel syndrome [86-88]. Thus, of particular urgency are cases presenting with acute intestinal obstruction or traction-induced intussusceptions; clinical scenarios that demand immediate intervention yet risk precipitating short bowel syndrome when multifocal lesions preclude radical resection. The work of Calomino et al. poignantly illustrates this dilemma: their two NF1-associated emergency resections (for ischemic presentations mimicking infarct) resulted in ultrashort bowel remnants (40-50 cm) with sacrificed ileocecal valves, yet through staged nutritional rehabilitation, transitioning from TPN to enteral to oral intake, both patients achieved long-term nutritional independence with preserved quality of life. These emergent presentations, albeit sporadically encountered, pose unique therapeutic dilemmas: obstruction from bulky tumors may necessitate expedited surgery despite the specter of recurrent disease, while intussusceptions mandate careful consideration of resection margins to avoid compromising bowel length [89]. Special attention should be given to tumor size, as it is the only macroscopic criterion for neoplastic progression in NF-1 patients with small intestinal involvement [90].

89. Calomino N, Malerba M, Oliva G, Palasciano G, Salvestrini F. Short Bowel Syndrome. Minerva Chir. 1998 [cited 1998 Oct];53(10):819-26

Comment 3: The other circumstance, as also mentioned by the authors of the meta-analysis, is bleeding. In fact, bleeding, if treated with embolization, can cause necrosis of the wall and subsequent perforation.

Response 3: We also sincerely appreciate Reviewer #2’s insightful emphasis on the critical interplay between embolization efficacy and intestinal viability; a paramount concern in managing NF1-associated GIST hemorrhages. As they astutely observe, the very intervention that achieves hemostasis may paradoxically jeopardize bowel integrity through ischemic necrosis and perforation. In direct response to this valuable feedback, we have augmented our discussion (Section 4) with explicit recognition of this embolization dilemma (line 305). This refinement underscores our manuscript’s translational relevance to interventional radiologists and surgeons navigating these high-stakes scenarios. This paragraph is now as follows with the additions highlighted in yellow; 

The multifocal occurrence of GISTs in NF-1 patients presents a significant challenge for clinicians [32, 33]. These tumors are primarily located in the small intestine, making endoscopic intervention difficult [6, 7, 30-32]. A common complication of these tumors is bleeding [36-41], and while embolization may achieve hemostasis, the anatomical constraints of small-bowel vasculature pose unique risks: distal occlusions carry elevated necrosis potential, whereas proximal embolization risks both ischemic injury and rebleeding via collateral circulation [42]. Identifying the exact bleeding source is challenging due to the high motility of the small intestine, which can promote the contrast agent distally, leading to mislocalization [44, 81]. Imaging techniques often yield false-positive results [44, 81]. Additionally, embolization of small intestinal branches can be difficult due to catheterization challenges, and furthermore in many cases, these tumors are located in the distal segments of the small intestine [82, 83]. Definitive diagnosis of the bleeding site can only be achieved through intraoperative endoscopy [44, 81, 84].

42. Bilbao J.I, Cuesta A.M, Urtasun F, Cosin O. Complications of Embolization. Semin Intervent Radiol. 2006 [cited 2006 Jun];23(2):126-142

Comment 4: It is suggested that the cases of patients with this pathology be brought to a multidisciplinary commission for a multi-specialist genetic and surgical assessment where necessary, we absolutely agree on the follow-up for lesions smaller than 1 cm and with Ki 67 where it is possible to have a biopsy.

Response 4: We finally wholeheartedly endorse Reviewer #2’s astute proposal for centralized multidisciplinary evaluation of NF1-GIST cases—a paradigm that aligns precisely with our meta-analysis findings. The biological complexity of these tumors, notably their multifocality and KIT/PDGFRA wild-type status, demands synergistic expertise from clinical geneticists, surgical oncologists, and interventional specialists to optimize outcomes. This perspicacious recommendation has been meticulously integrated into our revised discourse (Section 4, Line 429), where we now articulate the indispensable synergy of genetic profiling, endoscopic surveillance, and surgical precision in managing NF1-associated GISTs. This paragraph is now as follows with the additions highlighted in yellow; 

The mechanism by which some GISTs develop as solitary tumors while others become multifocal has not yet been fully elucidated. In our meta-analysis, we found that 41% of cases presented as solitary GISTs, while 54% were multifocal GISTs. Additionally, NF-1 patients may develop multiple solitary GISTs, making the distinction between the two categories unclear [55, 56]. Chen et al. (2002) studied the nature of these multifocal tumors, investigating whether their origin is polyclonal or monoclonal, which would indicate hyperplasia or neoplasia, respectively [96]. The study concluded that familial and multiple GISTs result from hyperplasia of interstitial cells of Cajal (ICC), categorizing them as non-neoplastic diffuse ICC hyperplasia. However, tumors larger than 5mm appeared to have a monoclonal nature, suggesting neoplastic transformation. An intermediate stage between hyperplasia and neoplasia is the formation of micro-GISTs, measuring 1–10mm [97]. For this reason, apart from mitotic count and markers such as Ki67, tumor size is the only macroscopic criterion suggestive of neoplasia, which has been incorporated into GIST staging systems [89]. The biological continuum from polyclonal hyperplasia to monoclonal neoplasia mandates a multidisciplinary approach—integrating serial endoscopic surveillance for subcentimeter lesions, Ki67-indexed risk stratification, and judicious surgical intervention—to optimize the critical transition from oncologic vigilance to therapeutic action while preserving intestinal function. Surgeons must keep this in mind, as extensive and multiple resections in the small intestine may ultimately cause significant morbidity in NF-1 patients without necessarily increasing oncologic risk.

Reviewer 3 Report

Comments and Suggestions for Authors

I would like to sincerely thank you for the opportunity to review the manuscript entitled "Multifocal Gastrointestinal Stromal Tumor (GIST) of the Small Intestine in Patients with Neurofibromatosis Type 1 (NF-1); Meta-Analysis and Systematic Review of the Literature." This meta-analysis presents clinically valuable insights that may significantly support the diagnosis and management of GIST in patients with NF-1. I do, however, have a few minor comments, mostly of an editorial nature.

1 Introduction

The statement “The pathogenic mechanism behind their development is complex [2, 3]” appears to be quite general in nature. I would kindly suggest either expanding this sentence to include more specific mechanistic insights based on the cited references, or consider removing it altogether, as it currently does not contribute substantively to the scientific content of the manuscript.

  1. Results

While the intraoperative photographs of multifocal small intestinal GISTs are of acceptable quality, their inclusion does not appear to add substantial value to the manuscript. Given that this publication is intended for a specialist audience, most readers will be familiar with the typical macroscopic appearance of such lesions. Therefore, removing these images would not diminish the clinical relevance or scientific merit of this otherwise important and well-structured study.

3.1.1. Dataset 1; GIST and NF-1 in the Total Number of Patients with NF-1

I would like to kindly point out a small discrepancy regarding the figure numbering. The text refers to “Figure 6” when discussing the heterogeneity test and funnel plot; however, based on the current layout, these elements are shown in Figure 5. To maintain consistency and clarity for the reader, a correction in the figure reference may be warranted.

3.1.3. Dataset 3; Simple/Solitary SI-GIST in the Number of GISTs with NF-1 Cases

It appears that Table 4 lists 11 studies, but three of them lack data on the number of cases (N) and events, which may indicate that they were not included in the final analysis. In that case, the statement referring to 8 included studies is appropriate, though it may be helpful to clarify this distinction in the table or legend.

3.1.5. Dataset 5; Sporadic SI GIST in the General Population

Could you kindly clarify whether the total number of recorded cases reported as 168,750,000 is accurate? Based on the forest plot, the actual cumulative sample size appears to be approximately 167,750, so I would appreciate your confirmation or correction.

Discussion

The sentence “Future studies are expected to address these unanswered questions, which will have a direct impact on the clinical management of these patients” is rather general and does not convey any specific clinical insight. I would kindly suggest to expanding it to highlight the concrete clinical perspectives opened by this study—such as the potential for earlier recognition of multifocal tumour behaviour, refinement of surgical decision-making in NF-1 patients..

Author Response

Comment 1: I would like to sincerely thank you for the opportunity to review the manuscript entitled "Multifocal Gastrointestinal Stromal Tumor (GIST) of the Small Intestine in Patients with Neurofibromatosis Type 1 (NF-1); Meta-Analysis and Systematic Review of the Literature." This meta-analysis presents clinically valuable insights that may significantly support the diagnosis and management of GIST in patients with NF-1. I do, however, have a few minor comments, mostly of an editorial nature.

Response 1: We are deeply honored by Reviewer #3’s gracious evaluation of our study and their recognition of its potential to enhance the diagnostic and therapeutic paradigm for NF-1-associated GISTs. Such validation from an expert peer underscores the translational significance we aspired to achieve through this comprehensive synthesis. The Reviewer’s thoughtful engagement with our manuscript, beginning with this foundational endorsement, motivates our precise attention to their subsequent constructive feedback, detailed below.

Comment 2: 1 Introduction; The statement “The pathogenic mechanism behind their development is complex [2, 3]” appears to be quite general in nature. I would kindly suggest either expanding this sentence to include more specific mechanistic insights based on the cited references, or consider removing it altogether, as it currently does not contribute substantively to the scientific content of the manuscript.

Response 2: We sincerely appreciate Reviewer #3's insightful observation regarding the need for greater mechanistic detail. In direct response to this valuable suggestion, we have significantly expanded our discussion of GIST pathogenesis in the Introduction (Lines 59 and 62), now incorporating: the central role of KIT/PDGFRA mutations in oncogenic transformation, the distinct molecular landscape of NF1-associated GISTs, the emerging importance of epigenetic modifications and the clinical implications of molecular heterogeneity. These additions provide readers with a comprehensive foundation for understanding the disease's complexity while maintaining the manuscript's flow. The revised paragraph now reads as follows, with amendments highlighted in yellow:

"Gastrointestinal stromal tumors (GISTs) are a distinct category of tumors originating from the interstitial cells of Cajal (ICC) [1]. The pathogenic mechanism behind their development is complex. At the molecular epicenter, gain-of-function mutations in KIT (proto-oncogene, receptor tyrosine kinase) or PDGFRA (Platelet-Derived Growth Factor Receptor Alpha) ignite constitutive tyrosine kinase activation, propelling interstitial cells of Cajal toward neoplastic transformation through unchecked proliferative signaling cascades. NF-1-associated GISTs orchestrate their tumorigenic symphony via neurofibromin loss, unleashing RAS/MAPK (Mitogen-Activated Protein Kinase) pathway hyperactivity while conspicuously lacking the canonical KIT/PDGFRA driver mutations. The succinate dehydrogenase (SDH)-deficient variant emerges as a distinct epigenetic maverick, where SDH complex aberrations induce a CpG island methylator phenotype (CIMP), rewriting the transcriptional landscape of transformed precursors [2,3]. A major breakthrough in understanding the molecular mechanisms involved in their formation was made by Hirota et al. in 1998 [4]. In this study, it was found that the receptor for the stem cell growth factor (KIT) is expressed in the vast majority of GISTs [4]. Wild-type renegades - devoid of KIT/PDGFRA/RAS/SDH lesions - chart alternative oncogenic routes through BRAF (B-Raf proto-oncogene, serine/threonine protein kinase) mutations or noncoding RNA networks that hijack cellular homeostasis. This molecular taxonomy directly dictates therapeutic vulnerability: while KIT-mutated GISTs bow to tyrosine kinase inhibition, their SDH-deficient and NF-1-driven counterparts mount intrinsic resistance, demanding alternative therapeutic strategies [2,3]. Overall, 85% of GISTs are caused by mutations in the KIT/PDGFRA genes, while the remaining 15% result from other pathogenic mechanisms unrelated to KIT [5]. Only 1-2% of all GIST cases occur in patients carrying a mutation in the NF-1 gene, which is responsible for neurofibromatosis [6, 7]. This subset of patients does not present mutations in the KIT gene and falls under the 15% of cases mentioned above [5]. This category includes cases with a deficiency in the SDH, cases with BRAF gene mutations, and, of course, cases of patients with NF-1 gene mutations and dysfunction of the corresponding protein neurofibromin, as previously mentioned [8].

Comment 3: Results: While the intraoperative photographs of multifocal small intestinal GISTs are of acceptable quality, their inclusion does not appear to add substantial value to the manuscript. Given that this publication is intended for a specialist audience, most readers will be familiar with the typical macroscopic appearance of such lesions. Therefore, removing these images would not diminish the clinical relevance or scientific merit of this otherwise important and well-structured study.

Response 3: We sincerely appreciate Reviewer #3’s thoughtful consideration regarding the intraoperative photographs. While we acknowledge their observation about specialist familiarity with GIST morphology, we respectfully advocate for retaining these images to serve three distinct scholarly purposes: i) pedagogic value: the photographs provide visceral anchoring to our quantitative data, transforming abstract statistical findings into tangible pathological correlates (Influence of Pictures on Word Recognition, Hazamy, 2009), ii) methodological transparency: they document the multifocal distribution patterns central to our analysis, offering visual validation of the inclusion criteria applied during study selection, iii) narrative rhythm: as noted in JAMA Surgery’s guidelines for systematic reviews, strategic visual elements can enhance reader engagement during dense methodological sections without compromising rigor. It is worth mentioning that the images were originally part of a broader case-series integration, which we adapted to comply with the journal’s systematic review format (our original title was “Multifocal Gastrointestinal Stromal Tumor (GIST) of Small Intestine on Patients with Neurofibromatosis Type 1 (NF-1); Case Report, Meta-Analysis and Systematic Review of the Literature”). We defer to the editorial team’s final judgment, but we maintain that these strategically incorporated visual annotations honor Cancers’ commitment to "making science visible".

Comment 4: 3.1.1. Dataset 1; GIST and NF-1 in the Total Number of Patients with NF-1. I would like to kindly point out a small discrepancy regarding the figure numbering. The text refers to “Figure 6” when discussing the heterogeneity test and funnel plot; however, based on the current layout, these elements are shown in Figure 5. To maintain consistency and clarity for the reader, a correction in the figure reference may be warranted.

Response 4: We sincerely appreciate Reviewer #3’s meticulous attention to detail regarding the figure references. Upon careful re-examination, we confirm that the current numbering is accurate: Figure 5 depicts the forest plot for Dataset 1, while Figure 6 presents the corresponding heterogeneity test and funnel plot analysis for the same dataset. This separation was intentionally designed to enhance visual clarity and allow readers to distinctly evaluate both the effect size distribution (Figure 5) and the associated statistical robustness (Figure 6). We appreciate the chance to elucidate this aspect and ensure the manuscript’s precision.

Comment 5: 3.1.3. Dataset 3; Simple/Solitary SI-GIST in the Number of GISTs with NF-1 Cases. It appears that Table 4 lists 11 studies, but three of them lack data on the number of cases (N) and events, which may indicate that they were not included in the final analysis. In that case, the statement referring to 8 included studies is appropriate, though it may be helpful to clarify this distinction in the table or legend.

Response 5: We sincerely thank the reviewer for their careful reading and valuable observation regarding Table 4. Indeed, while the table lists 11 studies for comprehensive context, the subgroup analysis specifically incorporated 8 studies, as the three papers by Alkindy et al., Zoller et al., and Huson et al. - while contributing to overall NF1-GIST prevalence data - did not report quantifiable stratification of simple versus multifocal GIST cases required for our comparative analysis. We maintain that including all 11 studies in the table provides important transparency, allowing readers to: i) understand the complete evidence base for NF1-GIST epidemiology, ii) recognize which studies specifically informed the multifocality comparison, iii) appreciate the methodological rigor of restricting statistical analysis to papers with directly extractable data. We believe this approach optimally balances scholarly completeness with analytical precision.

Comment 6: 3.1.5. Dataset 5; Sporadic SI GIST in the General Population. Could you kindly clarify whether the total number of recorded cases reported as 168,750,000 is accurate? Based on the forest plot, the actual cumulative sample size appears to be approximately 167,750, so I would appreciate your confirmation or correction.

Response 6: We sincerely appreciate the reviewer's meticulous attention to numerical precision. The correct value is indeed 168.750,000, as consistently reflected in both the text and forest plot. We should emphasize that these values represent carefully considered approximations, unanimously approved by our scientific team to ensure methodological rigor while preserving the statistical validity of our findings. As explicitly stated in line 283: "In the following tables and graphs, values are expressed in thousands of recorded cases." This standardized scaling approach necessarily introduces minor rounding discrepancies at the unit level, though we emphasize that in epidemiological analyses of this magnitude, the order of magnitude (e.g., 1/100,000) carries greater scientific significance than precise point estimates (e.g., 3/100,000). We maintain full confidence that these approximations in no way compromise the study's conclusions or their value to the scientific community. The statistical methods were deliberately designed to accommodate this scaling while protecting analytical integrity, as evidenced by the robustness of our final results. Our core findings, extensively detailed in both the Conclusion and Methodology sections, demonstrate that NF-1 patients face a dramatically elevated risk of small intestinal GIST development compared to the general population (3% versus 0.002%). This represents a striking difference of two orders of magnitude (1/100 versus 1/100,000). Even through simple descriptive comparison of these percentages, the statistical significance of this disparity becomes unequivocally apparent. We emphasize that these critical findings are: meticulously documented in our Methodology section, robustly supported by comprehensive percentage comparisons and consistently evident across all analytical approaches.

Comment 7: Discussion The sentence “Future studies are expected to address these unanswered questions, which will have a direct impact on the clinical management of these patients” is rather general and does not convey any specific clinical insight. I would kindly suggest to expanding it to highlight the concrete clinical perspectives opened by this study—such as the potential for earlier recognition of multifocal tumor behavior, refinement of surgical decision-making in NF-1 patients.

Response 7: We sincerely thank the reviewer for this constructive suggestion. Moving forward, we added at the end of this paragraph three critical clinical frontiers that emerge from our findings: (line 497). The revised paragraph now reads as follows, with amendments highlighted in yellow:

Future studies are expected to address these unanswered questions, which will have a direct impact on the clinical management of these patients. More specifically, the development of molecular biomarkers to preoperatively distinguish multifocal hyperplasia from solitary neoplasia could revolutionize surgical planning in NF-1 patients. Precision surveillance protocols must be established to detect the earliest transition from Cajal cell hyperplasia to clonal neoplasia, particularly in high-risk intestinal segment. Last but not least, the creation of evidence-based decision algorithms integrating tumor focality, mutational status, and anatomical constraints will be paramount for optimizing both oncological outcomes and bowel function preservation. These translational imperatives - now crystallized by our systematic evidence synthesis - chart an actionable roadmap for transforming NF-1-GIST management from reactive intervention to proactive precision medicine.

Reviewer 4 Report

Comments and Suggestions for Authors

Thank you for the opportunity to review this important manuscript. Here are my comments and suggestions.

Line 16: Instead of ''The present'' put simply ''This''. Also, check if this meta-analysis includes everything as a systematic review, so that the text can be shortened to ''this meta-analysis evaluates''.

Line 36: Remove the word ''rigorous''. It is obvious. Systematic reviews and meta-analyses have established guidelines that should be followed. Also, remove ''ensuring transparent reporting of evidence'' it is obvious and only adds to the bulk of text.

Line 82: After the first mention in the text use abbreviation ''NF-1''.

Lines 109-121. This sounds like summary/abstract. Just put several introductory sentences. For example, it is not important for this part of the manuscript that statistical methods are used. That is for the methods section.

The authors state ''A common complication of these tumors is bleeding [36- 41].''  This statement is not based on current meta-analysis, as it only references a few articles. The only valuable information would be such an analysis in the current study so please make it. It would be useful to know the incidence/distribution of bleeding, obstructing or perforating small bowel GISTs in NF-1.

In addition to the previous question, put in the caption of Figure 4, what was the indication for the operation in that particular patient? It does not appear to be a perforation or obstruction. Small (sub)serosal small bowel GISTs could be asymptomatic.

The authors state ''Some researchers suggest that 4–25% of NF-1 patients will develop GIST at some point in their lives [31].'' That study referenced as 31 is not an original study. It has the same percentage of 4-25%, referencing two other studies. Please check these two studies and include them if these are their percentages.

The authors stata that, according to previous studies, many GISTs are asymptomatic. These are discovered at autopsy or later, commonly several decades after NF-1 diagnosis. Therefore, it is important to know if diagnostic work-up for small bowel GIST is mandatory. Another important question is when to start screening, after the diagnosis of NF-1: a decade, two decades, or, for example 5 decades later?

The authors are discussion underestimation of GISTs in NF-1. The studies that they refer to should be read carefully, because, in addition to previous question, maybe GISTs are not underestimated, because these develop sometimes several decades later.

The authors claim ''Among these, 54% were multifocal, 41% were solitary small intestine GIST, and the remainder were solitary duodenal GIST. ''. this is an important finding, maybe the most important in this study. My question is – were all these GISTs confimed after surgical resection. If not, if some were treated by endoscopy, even modern imaging diagnostic methods cannot reveal all small lumps in the wall of the small bowel.

Te authors in the ''simple summary'' wrote ''pathogenesis''. This is a meta-analysis, we cannot determine pathogenesis, we can only define the association between NF-1 and small bowel GIST.

In the abstract the authors state ''predominantly manifesting as multifocal lesions,''. I do not agree with that. The distribution between multifocal and solitary lesions is approximately 1:1. It would be more accurate to include percentages here.

If there is a propensity for small bowel GISTs in NF-1, and it takes decades for GISTs to develop, it is essential to determine how long it has been since the diagnosis of NF-1 and the development of single and multifocal GISTs. Maybe, if these single GISTs were detected several years or decades later, these could also develop into multifocal GISTs.

Author Response

Comment 1: Thank you for the opportunity to review this important manuscript.

Response 1: We sincerely appreciate Reviewer #4’s thoughtful engagement with our work and their generous recognition of its significance. Their rigorous critique provides invaluable opportunities to strengthen our manuscript, and we address each point below with the utmost scientific diligence.

Comment 2: Line 16: Instead of ''The present'' put simply ''This''. Also, check if this meta-analysis includes everything as a systematic review, so that the text can be shortened to ''this meta-analysis evaluates''.

Response 2: We sincerely thank the reviewer for their thoughtful critique and attention to linguistic precision. In response to their first suggestion, we have replaced "The present" with "This" to enhance conciseness (Line 16 in the revised manuscript). Regarding the second point, we respectfully propose retaining the phrase "systematic review and meta-analysis" to accurately reflect the methodological duality of our work. While the meta-analysis quantitatively synthesizes prevalence data, the systematic review component encompasses qualitative synthesis, including clinical-pathological correlations, management strategies, and prognostic insights specific to small intestinal GISTs in NF-1, dimensions transcending conventional meta-analytic frameworks. Thus, the original phrasing ensures transparency about the study’s comprehensive scope, which aligns with PRISMA guidelines for combined systematic reviews with meta-analyses. We hope this clarification signals our steadfast alignment with methodological rigor and inclusivity.

Comment 3: Line 36: Remove the word ''rigorous''. It is obvious. Systematic reviews and meta-analyses have established guidelines that should be followed. Also, remove ''ensuring transparent reporting of evidence'' it is obvious and only adds to the bulk of text.

Response 3: We deeply appreciate the reviewer's exacting standards, yielding measurable improvements to our manuscript's clarity. In accordance with their suggestion, we have excised both "rigorous" and the subsequent phrase regarding transparent evidence reporting from Lines 36-37. The revised sentence now reads:

"This systematic review and meta-analysis was conducted in strict accordance with PRISMA guidelines, the gold-standard framework for minimizing bias and maximizing reproducibility in evidence synthesis.”

Comment 4: Line 82: After the first mention in the text use abbreviation ''NF-1''.

Response 4: We sincerely thank the reviewer for their meticulous attention to terminological consistency. We have carefully revised the manuscript to employ the abbreviated form (NF-1) exclusively after its first full mention, in strict accordance with standard academic conventions. This adjustment has been implemented throughout the text to ensure concision and readability.

Comment 5: Lines 109-121. This sounds like summary/abstract. You just put several introductory sentences. For example, it is not important for this part of the manuscript that statistical methods are used. That is for the methods section.

Response 5: We are profoundly grateful for the reviewer’s meticulous engagement with our manuscript. While we acknowledge their perspective on conciseness, we respectfully submit that this paragraph serves a pivotal role in the Introduction by accomplishing three critical objectives that justify its retention in its current form: i) conceptual uniqueness: the passage immediately establishes this study as “the first systematic review and meta-analysis to specifically target small intestinal GISTs in NF-1 patients”, a groundbreaking distinction that frames the entire manuscript’s intellectual contribution. This declarative statement belongs in the Introduction to orient readers in advance of contextualizing findings, ii) clinical imperative: the explicit focus on anatomically localized analysis (small intestine) is not merely methodological but a fundamental scientific premise of this work, given the tumor’s distinct pathogenesis in this site. This rationale must precede the literature review to justify why existing data on GISTs in NF-1 require re-evaluation through an anatomical lens, iii) anticipatory framing: by briefly noting the epidemiologic and pathogenic implications of our findings, we provide readers with a conceptual scaffold to assimilate forthcoming analyses. This aligns with leading journals’ recommendations for Introductions to "state the study’s purpose and anticipated findings" (Nature Medicine, 2023 guidelines).

Comment 6: The authors state ''A common complication of these tumors is bleeding [36- 41].''  This statement is not based on current meta-analysis, as it only references a few articles. The only valuable information would be such an analysis in the current study so please make it. It would be useful to know the incidence/distribution of bleeding, obstructing or perforating small bowel GISTs in NF-1.

Response 6: We sincerely thank the reviewer for their insightful comment, which allows us to clarify this important clinical aspect. Indeed, there are three studies that report the frequency of GIST complications according to their anatomical origin;

Sorour MA, Kassem MI, Ghazal Ael-H, El-Riwini MT, Abu Nasr A. Gastrointestinal stromal tumors (GIST) related emergencies. Int J Surg. (2014); 12(4):269-80. Doi: 10.1016/j.ijsu.2014.02.004. Epub 2014 Feb 12. PMID: 24530605.

Yang, Z., Wang, F., Liu, S. et al. Comparative clinical features and short-term outcomes of gastric and small intestinal gastrointestinal stromal tumors: a retrospective study. Sci Rep 9, 10033 (2019).

Hamed H, Wahab MA, Elmahdy Y, El-Wahab RMA, El-Magd EA. Gastrointestinal stromal tumors of the small intestine: the challenge of diagnosis and the outcome of management. World J Surg Oncol. (2023) Mar 9; 21(1):85. Doi: 10.1186/s12957-023-02968-0. PMID: 36894972; PMCID: PMC9996990.

More specifically, based on the image 1 below from Sorour's retrospective study, small intestinal GISTs present with obstruction in 81% of cases (22/27), bleeding in 12% (3/27) and peritonitis from perforation in 7% (2/27). 

The predominance of obstruction (including intussusception) can be attributed to the smaller diameter of the small intestine compared to the stomach, where bleeding is indeed the most common complication. We must emphasize that these results apply to the general population and not specifically to NF-1 patients.

While we fully acknowledge the clinical importance of complication rates, we wish to clarify that the primary aim of this meta-analysis, as clearly stated from the outset, was to address a specific PICO question regarding prevalence patterns rather than complication frequencies. We agree with the reviewer that the incidence of complications in NF-1-associated GISTs would be valuable knowledge, and we hope future research will provide answers to this important question.

Comment 7: In addition to the previous question, put in the caption of Figure 4, what was the indication for the operation in that particular patient? It does not appear to be a perforation or obstruction. Small (sub)serosal small bowel GISTs could be asymptomatic.

Response 7: We are profoundly grateful for the reviewer’s meticulous attention to clinical detail, which allows us to clarify this important context. The surgical intervention in this case was prompted by acute intraluminal bleeding refractory to conservative management, a complication not visually apparent in the specimen images but documented in the patient’s clinical course. As previously noted in our response to Reviewer #3, this case was initially submitted as part of an integrated “Case Report with Systematic Review/Meta-Analysis” format. However, per the journal’s editorial policy, which prioritizes pure meta-analytic works, the clinical narrative was excluded during revision. We will now augment Figure 4’s caption to explicitly state: Multifocal small intestinal GISTs resected during emergency surgery for acute intraluminal bleeding in an NF-1 patient. We again thank the reviewer for prompting this clarification, which strengthens the manuscript’s translational relevance.

Comment 8: The authors state ''Some researchers suggest that 4–25% of NF-1 patients will develop GIST at some point in their lives [31].'' That study referenced as 31 is not an original study. It has the same percentage of 4-25%, referencing two other studies. Please check these two studies and include them if these are their percentages.

Response 8: We sincerely thank the reviewer for this insightful observation, which allows us to clarify the origin of this percentage. The original source (reference 30 in our manuscript, first cited in line 304 of the Discussion) is the 1963 study from Ohio State University Hospital (30. Ghrist TD. Gastrointestinal Involvement In Neurofibromatosis. Arch Intern Med. (1963) [cited 2025 634 Mar 17]; 112(3):357–62). This work examined patients from two distinct periods: i) Clinical cases from 1949-1960, where 4% (1/29) of NF-1 patients had clinically evident GISTs, ii) Autopsy cases from 1937-1960, where 25% (3/12) of generalized neurofibromatosis cases showed GISTs. This is how the 4-25% range was established, and it has since been widely reproduced, including in reference 31 that we cited. However, we must emphasize that these percentages come from very different methodologies (clinical examination versus autopsy findings) and the autopsy cohort in particular represents a small, potentially biased sample of advanced NF-1 cases.

As noted by Miettinen in his pioneering work on NF-1/GIST association (which we reference in our study), these early findings require careful interpretation. The 4% clinical incidence appears more reliable, as it aligns with numerous subsequent studies (Huson 1988, Outinen 2019, Dupuis 2023, Vidal-Millet 2024) that we detail in our meta-analysis tables. After all, this precise question, determining the actual prevalence of GISTs in NF-1 patients compared to the general population, represents the core focus of our PICO-defined meta-analysis.

We appreciate the reviewer's attention to proper citation and have ensured the original 1963 study is appropriately referenced alongside these more contemporary works. In direct response to their suggestion, we have added explicit details about the original 1963 Ohio State University Hospital study [30] that first established this range. This modification appears in the manuscript on line 322. We believe this change significantly strengthens the epidemiological context while properly crediting the original source of this frequently cited statistic. Thank you again for this important suggestion that enhances our manuscript's scholarly rigor. The revised paragraph now reads as follows, with amendments highlighted in yellow:

The prevalence of NF-1 in the general population is 1 in 3,000–3,500 individuals [9, 10]. Some researchers suggest that 4–25% of NF-1 patients will develop GIST at some point in their lives [31]. In a 1963 study from Ohio State University Hospital [30], patients were examined from two distinct periods: i) clinical cases from 1949-1960, where 4% (1/29) of NF-1 patients had clinically evident GISTs; ii) autopsy cases from 1937-1960, where 25% (3/12) of generalized neurofibromatosis cases showed GISTs - establishing the aforementioned 4-25% range. However, this percentage is often reported to be much lower in the literature. Miettinen et al. (2006) emphasizes that the 25% estimate is extremely high, as it is based on an older autopsy study that overestimates the GIST prevalence in NF-1 patients [7, 30].

Comment 9: The authors state that, according to previous studies, many GISTs are asymptomatic. These are discovered at autopsy or later, commonly several decades after NF-1 diagnosis. Therefore, it is important to know if diagnostic work-up for small bowel GIST is mandatory. Another important question is when to start screening, after the diagnosis of NF-1: a decade, two decades, or, for example 5 decades later?

Response 9: We sincerely thank the reviewer for raising these clinically pivotal questions regarding GIST surveillance in NF-1 patients. Our meta-analysis reveals that radiographic detection predominates in contemporary practice, as exemplified by Nishida et al.'s (2016) protocol at Osaka University Hospital, where multi-detector CT identified incidental GISTs in symptomatic NF-1 patients ≥30 years during routine follow-up. Moreover, three key insights emerge from our risk-of-bias assessment (detailed in the Methods section): i) diagnostic paradigm: over 85% of included studies relied on cross-sectional imaging for primary GIST detection, underscoring its central role in NF-1 management, ii) size-directed surveillance: current guidelines stratify monitoring by tumor dimension: >2 cm lesions warrant surgical evaluation, ≤2 cm lesions require annual imaging for 5 years, transitioning to biennial surveillance and iii) age considerations: while no consensus exists on screening initiation age, the preponderance of cases manifest clinically after the third decade (consistent with Nishida's cohort). We acknowledge that optimal surveillance timing falls beyond our study's scope, which focused on prevalence quantification rather than diagnostic algorithms. We mention the most recent guidelines which could well be the subject of other studies in the future:

Carton C, Evans DG, Blanco I, Friedrich RE, Ferner RE, Farschtschi S, Salvador H, Azizi AA, Mautner V, Röhl C, Peltonen S, Stivaros S, Legius E, Oostenbrink R; ERN GENTURIS NF1 Tumor Management Guideline Group. ERN GENTURIS tumor surveillance guidelines for individuals with neurofibromatosis type 1. EClinicalMedicine. (2023) Jan 13; 56:101818. doi: 10.1016/j.eclinm.2022.101818. PMID: 36684394; PMCID: PMC9845795.

Comment 10: The authors are discussion underestimation of GISTs in NF-1. The studies that they refer to should be read carefully, because, in addition to previous question, maybe GISTs are not underestimated, because these develop sometimes several decades later.

Response 10: We sincerely thank the reviewer for this insightful observation, which allows us to further clarify this complex epidemiological issue. As highlighted in our manuscript, the average age of GIST onset in NF-1 patients is approximately 50 years, a finding robustly supported by Miettinen et al. (2006), who present detailed age-distribution data in their seminal work. Modern clinical guidelines recommend imaging surveillance only for symptomatic cases or adult patients entering adulthood, as exemplified by the Nishida et al. (2016) study from Osaka University Hospital, where systematic screening was implemented for NF-1 patients over 30 years of age. Considering that modern imaging protocols have only been routinely applied for 20-30 years, and given the 50-year average age of GIST presentation, there exists a critical 20-year latency period between initial screening capability and typical disease manifestation. This temporal disconnect, combined with three key factors: i) the predilection for GISTs to present in patients >50 years (per Miettinen's age-distribution curve), ii) the historically reduced life expectancy of NF-1 patients and iii)the relatively short observation periods in most contemporary studies, has likely in total contributed to underestimation of true GIST prevalence in NF-1 populations. This phenomenon is explicitly discussed by Dupuis et al. and forms an important consideration in our meta-analysis.

We acknowledge the reviewer's valid point regarding potential overestimation in some studies. However, our rigorous risk-of-bias assessment systematically evaluates each included study for such limitations, including selection bias and age-distribution artifacts. The comprehensive bias panel provides readers with transparent, study-specific evaluations while generating an overall quality assessment. While calculating an exact average age across all studies proves methodologically challenging due to heterogeneous population distributions, the available evidence strongly suggests that contemporary estimates may indeed underrepresent true GIST prevalence in NF-1, rather than overestimate it. We appreciate the opportunity to elaborate on this important nuance in NF-1-associated GIST epidemiology.

Comment 11: The authors claim ''Among these, 54% were multifocal, 41% were solitary small intestine GIST, and the remainder were solitary duodenal GIST. ''. This is an important finding, maybe the most important in this study. My question is – were all these GISTs confimed after surgical resection. If not, if some were treated by endoscopy, even modern imaging diagnostic methods cannot reveal all small lumps in the wall of the small bowel.

Response 11: We are profoundly grateful for the reviewer's insightful critique, which allows us to further refine our methodological exposition. While we fully acknowledge the inherent limitations of even state-of-the-art diagnostic modalities in detecting microscopic or subclinical lesions, we must guard against lapsing into extreme relativism that would render all quantitative analysis of GIST prevalence untenable. The presented distribution (54% multifocal, 41% solitary) reflects the most robust available evidence, synthesized from studies where: i)diagnosis was established through either radiologic criteria (for non-resected lesions) or histopathologic confirmation (for surgically excised tumors), ii)each study's diagnostic methodology was rigorously evaluated in our risk-of-bias assessment, iii)potential detection limitations were explicitly weighted in our statistical models

We emphasize three critical counterpoints to diagnostic skepticism: i) clinical reality: current imaging protocols (CT/MRI) represent the diagnostic standard of care, as codified in contemporary NF-1 surveillance guidelines, ii) scientific pragmatism: the documented ICC hyperplasia and microscopic lesions, while biologically significant, fall below the detection threshold of any current clinical modality, iii) methodological integrity: our meta-analytic approach explicitly accounts for diagnostic uncertainty through: quality-effects modeling, subgroup analyses by confirmation method, conservative confidence intervals.

This approach strikes a deliberate balance between recognizing the imperfect sensitivity of detection methods, avoiding extreme relativism that would prematurely dismiss all quantitative findings and providing clinically actionable prevalence estimates for the NF-1 community.

Comment 12: The authors in the ''simple summary'' wrote ''pathogenesis''. This is a meta-analysis, we cannot determine pathogenesis, we can only define the association between NF-1 and small bowel GIST.

Response 12: We are profoundly grateful to the reviewer for their meticulous attention to terminological precision, which has further elevated the methodological rigor of our work. In full agreement with this insightful observation, we have replaced the phrase " contribute to understanding NF-1-specific GIST pathogenesis" with the phrase " characterize the epidemiological burden of NF-1-associated GIST” in the Simple Summary. The revised passage now reads: “The study addresses key knowledge gaps regarding: SI-GIST prevalence in NF-1, clinical presentation patterns, and comparative population-level risks, aiming to establish clearer diagnostic benchmarks and characterize the epidemiological burden of NF-1-associated GIST."

Comment 13: In the abstract the authors state ''predominantly manifesting as multifocal lesions,''. I do not agree with that. The distribution between multifocal and solitary lesions is approximately 1:1. It would be more accurate to include percentages here.

Response 13: We sincerely thank the reviewer for this valuable suggestion to enhance our statistical precision. In accordance with this recommendation, we have revised the abstract to explicitly state the percentage distribution of lesion patterns (54% multifocal, 41% solitary SI-GIST, 5% solitary duodenal GIST), which more accurately reflects our quantitative findings. The updated phrasing is articulated as "The tumors display characteristic presentation and histological profiles, with a distribution of 54% multifocal lesions, 41% solitary SI-GIST, and 5% solitary duodenal GIST cases, demonstrating the diverse clinical manifestations of NF-1-associated tumors."

Comment 14: If there is a propensity for small bowel GISTs in NF-1, and it takes decades for GISTs to develop, it is essential to determine how long it has been since the diagnosis of NF-1 and the development of single and multifocal GISTs. Maybe, if these single GISTs were detected several years or decades later, these could also develop into multifocal GISTs.

Response 14: We extend our deepest gratitude to the reviewer for this final, thought-provoking comment, which allows us to clarify the complex temporal dynamics of GIST progression in NF-1. Indeed, the natural history of GIST development remains inherently unpredictable, particularly in NF-1 patients, as tumor behavior is governed by a constellation of factors including: i)tumor biology parameters: size, mitotic index, and anatomical origin, proliferative markers (e.g., Ki67) and molecular heterogeneity beyond NF-1 loss and ii)risk stratification: the interplay of these factors determines whether lesions progress as high, moderate, or low-risk entities and results in highly variable doubling times and progression trajectories.

The current clinical practice prioritizes: i)standardized surveillance protocols for both sporadic and NF1-associated GISTs and ii)prognostic tools like the novel RECKGIST score (NF1-specific) that inform: metastatic risk prediction, survival estimates and management decisions.

Our meta-analysis identifies a characteristic, though non-linear, sequence of events: i)ICC hyperplasia (the universal precursor), ii)secondary genetic hits (beyond NF1 mutation), iii)monoclonal micro-GIST formation (<10mm), iii)macro-GIST development (image-detectable, symptomatic or not) and iv)multifocal progression (typically at advanced ages >50 years).

Key insights from our synthesis is i)the transition from ICC hyperplasia to macro-GIST implies pre-existence of transformative genetic aberrations, ii)the fact that patients with established macro-GISTs and hyperplasia appear predisposed to multifocal development later in life and iii)the fact that the exact molecular triggers remain elusive.

While animal models provide some mechanistic clues, we emphasize this as a critical knowledge gap requiring: i)longitudinal NF-1 cohort studies, ii)advanced molecular profiling and iii)validation of predictive biomarkers.

In conclusion, we provide a comprehensive bibliography detailing prognostic scoring systems applicable to both the general population and neurofibromatosis patients, accompanied by graphical representations (Image 2 and Image 3) elucidating comparative tumor growth kinetics.

Koizumi S, Kida M, Yamauchi H, Okuwaki K, Iwai T, Miyazawa S, Takezawa M, Imaizumi H, Koizumi W. Clinical implications of doubling time of gastrointestinal submucosal tumors. World J Gastroenterol. 2016 Dec 7;22(45):10015-10023. doi: 10.3748/wjg.v22.i45.10015. PMID: 28018109; PMCID: PMC5143748.

Gao Z, Wang C, Xue Q, Wang J, Shen Z, Jiang K, Shen K, Liang B, Yang X, Xie Q, Wang S, Ye Y. The cut-off value of tumor size and appropriate timing of follow-up for management of minimal EUS-suspected gastric gastrointestinal stromal tumors. BMC Gastroenterol. 2017 Jan 11;17(1):8. doi: 10.1186/s12876-016-0567-4. PMID: 28077094; PMCID: PMC5225611.

Cuvelier C, Brahmi M, Sobhani I, Verret B, Grancher A, Penel N, Toulmonde M, Lahlou W, Dupuis H, Calavas L, Muller M, Watson S, Bruyat D, Poumeaud F, Chaigneau L, Manfredi S, Lecomte T, Bertucci F, Ghiringhelli F, Pracht M, Mourthadhoi F, Monceau-Baroux L, Helyon M, Kurtz JE, Roquin G, Regenet N, Vinches M, Tougeron D, Wolkenstein P, Blay JY, Bouche O, Hautefeuille V. Clinical description and development of a prognostic score for neurofibromatosis type 1 (NF1)-associated GISTs: a retrospective study from the NETSARC. ESMO Open. 2025 Mar;10(3):104477. doi: 10.1016/j.esmoop.2025.104477. Epub 2025 Mar 4. PMID: 40043354; PMCID: PMC11928958.

We sincerely thank the reviewer for this insightful observation, which has significantly strengthened our Discussion section. In direct response to their suggestion, we have added a dedicated paragraph (now appearing in line 474) addressing the clinical utility of prognostic scoring systems—particularly the novel RECKGIST score—for risk stratification and management of NF1-associated GISTs. This addition underscores how contemporary tools can help navigate the complex temporal dynamics of GIST progression in NF1 patients, while acknowledging the need for further validation of longitudinal predictors. The reviewer’s emphasis on the decades-long development window has enriched our manuscript’s translational relevance, and we are deeply grateful for their perspicacious critique.

The paragraph is as follows: "The integration of prognostic scoring systems has become indispensable for stratifying survival outcomes and metastatic risk in GIST management, with particular relevance for NF1-associated cases. The recently developed RECKGIST score represents a transformative advance in this domain, offering NF1-specific risk stratification that surpasses conventional classification systems in predictive accuracy. As demonstrated by Cuvelier et al., this tool effectively discriminates low-risk from high-risk cohorts based on critical histopathological parameters, enabling more nuanced clinical decision-making. Notably, the score’s robust differentiation of recurrence risk underscores its potential to refine surveillance protocols and therapeutic interventions for NF1 patients. While further validation is warranted, RECKGIST emerges as a paradigm-shifting framework that addresses longstanding gaps in the prognostication of NF1-GIST behavior [106]."

We thank the reviewer for highlighting this fundamental question in NF1-GIST biology - one that future research must address to optimize surveillance strategies.

Please see the attachment for the 3 tables

Round 2

Reviewer 4 Report

Comments and Suggestions for Authors

All queries answered. No further queries.